# Ambient Dataloops:
# Generative Models for Dataset Refinement

**Adrian Rodriguez-Munoz** [1]  **William Daspit** [2]  **Adam Klivans** [2]  **Antonio Torralba** [1]  **Constantinos Daskalakis** [1]
**Giannis Daras** [1]

## Abstract

We propose Ambient Dataloops, an iterative framework for refining datasets that makes it easier for diffusion models to learn the underlying data distribution. Modern datasets contain samples of highly varying quality, and training directly on such heterogeneous data often yields suboptimal models. We propose a dataset-model co-evolution process; at each iteration of our method, the dataset becomes progressively higher quality, and the model improves accordingly. To avoid destructive self-consuming loops, at each generation, we treat the synthetically improved samples as noisy, but at a slightly lower noisy level than the previous iteration, and we use Ambient Diffusion techniques for learning under corruption. Empirically, Ambient Dataloops achieve state-of-the-art performance in unconditional and text-conditional image generation and de novo protein design. We further provide a theoretical justification for the proposed framework that captures the benefits of the data looping procedure.

## 1. Introduction

Much of the recent progress in generative modeling is attributed to the existence of large-scale, high-quality datasets. Indeed, modern generative models have an appetite for data that is becoming increasingly hard to fulfill (Goyal et al., 2024; Kaplan et al., 2020; Saharia et al., 2022; Hoffmann et al., 2022; Henighan et al., 2020). That triggers the formation of datasets that include any points that are available for training, including synthetic and out-of-distribution data, and naturally, these datasets contain samples of various qualities. The lower-quality parts of the training data are often

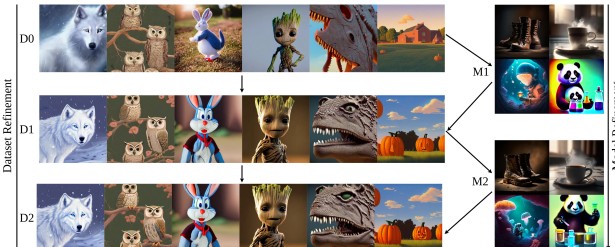

*Figure 1.* **Dataset and model evolution across loops of our method.** $D_0$ shows synthetically generated images from DiffusionDB (Wang et al., 2022), a dataset used for text-to-image generative modeling. These images have artifacts due to learning errors of the underlying model. We train a model on this dataset, $M_1$, that we use to improve its own training set, leading to a "restored" dataset $D_1$. Successive iterations of this process lead to a co-evolution of both the model and the dataset – see dataset $D_2$ and model $M_1$ respectively. We avoid catastrophic self-consuming loops by accounting for learning errors at each iteration using Ambient Diffusion (Daras et al., 2025c; 2023) techniques for learning from imperfect data.

removed through various filtering techniques (Gadre et al., 2023; Li et al., 2024), either from the beginning of the training or in some intermediate training stage (Sehwag et al., 2025). This approach is optimal when the bottleneck is the computational budget for training, since it is better to allocate the limited compute to the higher-quality training points (Goyal et al., 2024; Hoffmann et al., 2022). However, when the issue is not computational budget, but availability of data, filtering increases quality but comes at the cost of reduced diversity in the generated outputs (Somepalli et al., 2023a;b; Daras et al., 2024; Prabhudesai et al., 2025; Carlini et al., 2023)

Post-training, diffusion generative models often undergo refinements of all sorts to sample faster (Salimans & Ho, 2022; Song et al., 2023), become aligned with reward models (Domingo-Enrich et al., 2024; Black et al., 2023), or reduce their parameter count (Meng et al., 2023). The noisy dataset that was used to train the model remains, on the contrary, static. We hence ask: *Is it possible to use a model trained on a noisy set to improve the set that it was trained on?*

We propose a dataset-model co-evolution process, termed **Ambient Dataloops**. At each iteration of this process, we

[1]Department of Electrical Engineering and Computer Science, Massachusetts Institute of Technology [2]Department of Computer Science, University of Texas at Austin. Correspondence to: Giannis Daras <gdaras@mit.edu>.

*Proceedings of the $43^{rd}$ International Conference on Machine Learning*, Seoul, South Korea. PMLR 306, 2026. Copyright 2026 by the author(s).

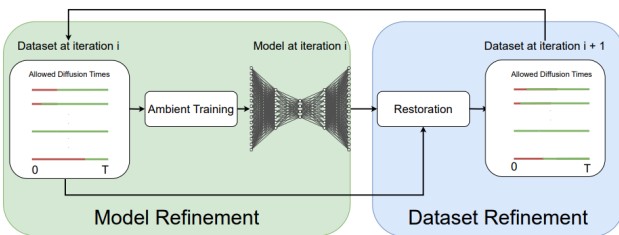

*Figure 2.* **Illustration of the Ambient Dataloops framework.** At each loop, we are given points that can be used to train the diffusion model at certain noise levels. We train a model on this noisy dataset using Ambient Diffusion (green), and then we use it to improve the dataset through posterior sampling (blue).

start with a noisy dataset, we use it to train a diffusion model, and then we use the trained model to **gradually** denoise the original dataset. We illustrate some results of this process for a text-conditional model in Figure 1. The concept of iterative dataset refinement has appeared in other contexts in the sampling from generative models literature (Akhound-Sadegh et al., 2024; Woo & Ahn, 2024; Denker et al., 2025). For example, the authors of (Woo & Ahn, 2024) use this idea to sample from unnormalized densities while the authors of (Denker et al., 2025) use it to sample from tilted measures using a reward model. In our work, we iteratively denoise a given noisy dataset by co-evolving the model being trained and the samples it is trained on. We avoid catastrophic, self-consuming loops, observed in prior works (Alemohammad et al., 2024a; Shumailov et al., 2024; Hataya et al., 2023; Martínez et al., 2023; Padmakumar & He, 2024; Seddik et al., 2024; Dohmatob et al., 2024) when training on self-generated outputs, by only slightly denoising the dataset each time and by performing corruption-aware diffusion training (e.g. as in Ambient Diffusion (Daras et al., 2025c;b; 2023)). The latter is used to account for errors that happen during the denoising process of the previous round and avoid propagating these errors to the next iteration. Experimentally, Ambient Dataloops consistently outperforms prior work on learning from corrupted data in both controlled settings, as well as in real datasets, including text-conditional models trained on dozens of millions of samples and generative models for protein structures. We further provide theoretical justification for the potential effectiveness of the approach in settings where the initial score estimation is sufficiently accurate.

## 2. Background and Related Work

**Diffusion Models.** Diffusion modeling (Sohl-Dickstein et al., 2015; Ho et al., 2020; Song & Ermon, 2019; Song et al., 2021) is one of the most prominent frameworks for learning high-dimensional, complex, continuous distributions. The main algorithmic idea is to consider not only the target density, which we will denote with $p_0$, but a family of intermediate distributions, $p_t = p_0 \circledast \mathcal{N}(0, \sigma(t)^2 I)$, where $\sigma(t)$ is an increasing function and $t$ is a contin-

uous variable in $[0, T]$ (for some big constant $T$) representing the diffusion time. We denote with $X_0$ the R.V. sampled according to the target density $p_0$ and similarly $X_t = X_0 + \sigma(t)Z$, $Z \sim \mathcal{N}(0, \sigma(t)^2 I)$ the R.V. sampled according to $p_t$. During training, the object of interest is the best $l_2$ denoiser for each one of these intermediate densities, i.e. the conditional expectation of the clean sample given a noisy observation, $\mathbb{E}[X_0|X_t = \cdot]$. The latter is typically optimized with the following objective:

$$J(\theta) = \mathbb{E}_{t \in \mathcal{U}[0,T]} \mathbb{E}_{X_0} \mathbb{E}_{X_t|X_0,t} \left[ ||h_\theta(X_t, t) - X_0||^2 \right]. \quad (1)$$

For a sufficiently rich parametrization family, the minimizer of this objective is indeed the conditional expectation, i.e. $h_{\theta^*}(\cdot, t) = \mathbb{E}[X_0|X_t = \cdot]$. The latter is connected to the score-function $\nabla \log p_t(\cdot)$ through Tweedie's formula (Tweedie, 1957; Efron, 2011) and it can be used to sample according to a diffusion process (Song et al., 2021; Anderson, 1982).

**Finite datasets and imperfect data.** In practice, we don't have access to infinite samples from $p_0$ but to a finite number, denote $n_1$. When $n_1$ is small, diffusion models often memorize their training set and learn the empirical distribution $\hat{p}_0$ (Shah et al., 2025; Daras et al., 2024; Somepalli et al., 2023a;b; Carlini et al., 2023; Kamb & Ganguli, 2025; Kadkhodaie et al., 2024). This pathological behavior, known as model collapse, has been studied in as series of recent works (Fu et al., 2025; Shi et al., 2025; Cui et al., 2025).

One way to increase the sample size and improve generalization is to incorporate low-quality or out-of-distribution data that is usually cheaper and more widely available. This occurs naturally in many datasets or can be collected (e.g. data scraping, synthetic data from other models, etc). To avoid hurting the generation quality or biasing the distribution, it is crucial to account for the corruption of this additional data during the training of the diffusion model. Over the past few years, there have been numerous proposed methods for training generative models with imperfect data (Bora et al., 2018; Daras et al., 2023; 2024; 2025a;c;b; Aali et al., 2023; 2025; Lu et al., 2025; Kelkar et al., 2024; Rozet et al., 2024; Bai et al., 2025; Zhang et al., 2026; Tewari et al., 2023; Liu et al., 2025; Alemohammad et al., 2024b). The majority of these works make some assumption about the nature of the degradation in the given data, which is limiting if we want to apply these datasets to Web-scale real datasets that have samples of various qualities and unknown corruption types.

Daras et al. (2025b;c) propose an approach for dealing with data of various qualities without an explicit degradation model. The central idea is that the distance between any two distributions, $p_0$ and $q_0$, contracts with the introduction of noise. In this work, $q_0$ is the distribution obtained by sampling from $p_0$ but via an unknown noisy measurement process. For a sufficiently high amount of noise, $t_n$,

the distributions $p_{t_n}$ and $q_{t_n}$ approximate well each other. Hence, for noise levels $t \geq t_n$, we can use samples from a low-quality or out-of-distribution data-source $q_0$ to increase the pool size of available data for a small distribution bias penalty. Daras et al. (2025c) analyze this bias-variance trade-off and provide rigorous ways for deciding the threshold $t_n$ beyond which it is beneficial to incorporate $q_0$ data. After annotation, each sample from $Y_0 \sim q_0$ is mapped to its noisy version $Y_{t_n}$ and the problem amounts to training a diffusion with a mixture of clean data (from $p_0$) and samples corrupted with additive Gaussian noise (that are well approximated as coming from $p_{t_n}$). For details see Appendix Section F.

The reduction of the problem to the additive noise case enables the leveraging of well-developed statistical tools for learning from noisy data (Stein, 1981; Lehtinen et al., 2018; Moran et al., 2020; Daras et al., 2024; 2025a). In particular, it is possible to learn the conditional expectation of the clean samples with noisy targets. In the most general form, we are given access to a dataset $\mathcal{D}$ where each sample has a known noise level $t_i$ and can be used for diffusion times in $[t_i, T]$. The two extremes are $t_i = 0$ (clean sample, used everywhere) and $t_i = T$ (filtering, the sample is not used at all). Daras et al. (2024) establish that for $\alpha(t, t_i) = \frac{\sigma^2(t) - \sigma^2(t_i)}{\sigma^2(t)}$, given enough data, the following objective has the same minimizer as equation 1, but it does so without having access to clean targets:

$$J_{\text{ambient-o}}(\theta) = \mathbb{E}_{t \in \mathcal{U}[0,T]} \sum_{i:t_i < t} \mathbb{E}_{x_t | x_{t_i}}$$

$$\left[ \left\| \alpha(t, t_i) h_\theta(x_t, t) + (1 - \alpha(t, t_i)) x_t - x_{t_i} \right\|^2 \right], \quad (2)$$

## 3. Method

**Problem Setting.** We study exactly the same problem as Daras et al. (2025c;b;a; 2024); in particular, we assume that we have access to a dataset of samples $\mathcal{D}_0 = \{(x_{t_i}, t_i)\}_{i=1}^N$ where each sample $x_{t_i}$ is (at least approximated as) being sampled from a density $p_{t_i}$. As explained in Section 2, this dataset is typically formed by starting with a dataset that contains some clean samples and some samples of unknown types and then adding the appropriate amount of noise to the corrupted samples to make them look approximately as clean samples corrupted with additive noise. In Section 5, we experiment with such transformed datasets. For the simplicity of the presentation, we avoid detailing this procedure in the main text, but we provide all the necessary information in Appendix Section F. We also refer the interested reader to (Daras et al., 2025c) for more details about how this initial reduction from arbitrary degradations to the additive Gaussian noise case can be performed.

**Algorithm.** Our method is summarized in Figure 2. It iterates between two steps, for $l = 1, \ldots$:

◇ **Model Training.** At this step, we take the training set $\mathcal{D}^{(l-1)}$ and then we train a new model on this dataset. Since the dataset has noisy data, we use the training objective of Equation 2. This is the standard step performed in prior work, e.g., see (Daras et al., 2025c;b;a; 2024).

◇ **Dataset Restoration.** At the end of the model training, we have a model $h_{\theta^{(l)}}$. Our method uses this network to *denoise* the *original dataset*. In particular, we perform posterior sampling $X_{t_i/2^l} \sim p_{\theta^{(l)}, t_i/2^l}(\cdot | x_{t_i}, t_i)$ and add $(X_{t_i/2^l}, t_i/2^l)$ to a new dataset $\mathcal{D}^{(l)}$. Simply put, this procedure synthetically reduces the noise level of the original dataset by denoising from $t$ to $t/2^l$ using the best prior model available at iteration $l$. The constant 2 effectively controls the amount of progress we expect each iteration of this algorithm to achieve, and in practice, it can be tuned as a hyperparameter.

A complete description of the algorithm is provided in Algorithm 1.

**Discussion.** The crux of this algorithm is dataset refinement; at each loop, we use the best model we have to improve the dataset by reducing the amount of noise in its samples. The resulting dataset can be used for a new training and so forth. As we run more loops, the model becomes better, and hence we take bigger denoising steps. We provide an overview of the approach in Figure 2.

**Comparison with Ambient Diffusion Omni and Expectation-Maximization-based works.** In (Daras et al., 2025c), the low-quality parts of the dataset are getting noised to a particular diffusion time, and they can only be used during training for times that correspond to higher noise than the noise level they got mapped to. This is the starting point of our method (loop 0). However, after a successful training, we can now use the model to partially denoise these original noisy points and perform a new training on the resultant dataset.

The concept of gradually improving a corrupted dataset using a diffusion prior has been explored previously in (Bai et al., 2024; Rozet et al., 2024; Hosseintabar et al., 2025; Modi et al., 2025). There are three critical differences between our framework and these works. Firstly, in Ambient Dataloops, the corruption process is *unknown* and in the first iteration we add noise on top of the corrupted samples to approximately reduce the problem to the Gaussian corruption case (as in (Daras et al., 2025c)). Second, contrary to prior work, we do not fully denoise the corrupted samples with the generative prior. This leads to computational benefits (as we do not need to run the full reverse process),

but more importantly, it accounts for the fact that the model is not perfect and mitigates propagation of learning errors throughout the refinement rounds. Finally, Dataloops uses the Ambient loss function of equation 2 instead of the normal score-matching loss used in prior works, since for the corrupted data we do not have clean targets.

**Potential limitations.** The idea of dataset refinement, despite being natural, has three issues. First, it seems to be violating the data processing inequality; information cannot be created out of thin air, and hence any processing of the original data cannot have more information for the underlying distribution than the original dataset. While this is true, it is important to consider that the first training might be suboptimal due to failures of the optimization process (e.g., gradient descent getting stuck in a local minimum). Hence, dataset refinement can be thought of as a reorganization of the original information in a way that facilitates learning and creates a better optimization landscape.

Another challenge for our method is that we train on synthetic data. Several recent works have shown that naive training on synthetic data leads to catastrophic self-confusing loops and mode collapse (Alemohammad et al., 2024a; Shumailov et al., 2024; Hataya et al., 2023; Martínez et al., 2023; Padmakumar & He, 2024; Seddik et al., 2024; Dohmatob et al., 2024). Our key idea to mitigate this issue is to treat the restorations as *noisy* data as well, just at a smaller noise level compared to where the restoration started. In particular, we do not run the full posterior sampling algorithm; we early stop the generation process at time $t_i/2^l$ at each round $l$. Prior work has shown that the catastrophic self-consuming loops can be avoided using a *verifier* that assesses the quality of the generations (Ferbach et al., 2024; Feng et al., 2024; Zhang et al., 2024). The gradual denoising and the finite number of rounds in our algorithm have a similar effect. Tuning the number of rounds wisely prevents the model from attempting to denoise the dataset at a level beyond what's possible using the available training set. Naturally, tuning this parameter in practice is not straightforward, and we provide ablations of miscalibration in our experiments. We clarify that miscalibrating either the number of loops or the rate at which we denoise the dataset at each loop can lead to performance deterioration.

The last issue associated with our approach has to do with the associated computational requirements. At each round, we have to restore the whole dataset and then fine-tune the model, leading to an increase in the training cost. Indeed, our method is useful when data, not compute, is the bottleneck. If there is more data available, it is always better to use it as fresh samples reveal more about the underlying distribution (Goyal et al., 2024) and there is no need to perform the expensive restoration step that our method requires. However, if there is lack of data, our framework is useful

as it attempts to extract as much utility as possible from the given training set.

## 4. Theoretical Modeling

In this section, we study the theoretical aspects of the proposed method. We consider a stylized setting and version of the algorithm, arguing that if the score function is sufficiently well-approximated after the first iteration, then performing a *dataset refinement* step can improve the estimation error.

**Setting.** For the purposes of the theoretical analysis, we adopt the theoretical setting from Ambient Omni (Daras et al., 2025c), and identify conditions under which *dataset looping* is beneficial.

In the description of our method, we assumed that we have access to a dataset $\mathcal{D}_0 = \{(x_{t_i}, t_i)\}_{i=1}^N$, where each datapoint comes with a threshold time $t_i$ indicating that we will use it to estimate scores for diffusion times $t \geq t_i$. As discussed earlier, the way those samples and associated threshold times came about is as follows: Some are samples from the target distribution $p_0$, and all these samples are assigned a threshold time 0. Then there are samples from distributions different from $p_0$. If some sample was sampled from some distribution $q_0$, we would add to it noise sampled from $\mathcal{N}(0, \sigma_{t_i}^2 I)$ and assign to the resulting noised sample threshold time $t_i$, where the choice of $t_i$ depends on the distance between $p_0$ and $q_0$. The choice would be such that $p_{t_i}$ and $q_{t_i}$ are sufficiently close that for $t \geq t_i$ we prefer to include this sample in estimating scores versus not using it. Choosing those times correctly is complex, but the theoretical analysis in Ambient Omni provides us guidance for how to choose these times, in the case where all samples either come from $p_0$ or from $q_0$, as described below. So let us stick to this case for our analysis here as well.

In particular, we are given $n_1$ samples from a target distribution $p_0$ that we want to learn to generate. We assume that $p_0$ is supported on $[0, 1]$ and is $\lambda_1$-Lipschitz. We are also given $n_2$ samples from a distribution $q_0$, which is not the target distribution, and may have some distance from $p_0$. We assume that $q_0$ is $\lambda_2$-Lipschitz. We want to train a diffusion model to sample $p_0$, so we need to learn the score functions of all distributions $p_t = p_0 \circledast \mathcal{N}(0, \sigma_t^2 I)$. Given our $n_1$ i.i.d. samples from $p_0$ we can create $n_1$ i.i.d. samples from $p_t$. Given our $n_2$ i.i.d. samples from $q_0$ we can also create $n_2$ i.i.d. samples from $q_t = q_0 \circledast \mathcal{N}(0, \sigma_t^2 I)$, but again $q_t$ is different from $p_t$. The observation that Daras et al. (2025c) leverage is that $q_t$ is closer to $p_t$ than $q_0$ is to $p_0$ because convolution with a Gaussian distribution contracts distances.

Because of this contraction, it could be that for sufficiently large $t$'s (a.k.a. $\sigma_t$'s), we are better off including the $n_2$ (bi-

ased) samples from $q_t$ to estimate $p_t$ rather than only using the unbiased samples from $p_t$. Indeed this is what is shown by Daras et al. (2025c) in Ambient Omni, as discussed below.

**Prior results.** For any diffusion time $t$, Daras et al. (2025c) compare the accuracy attained by the following algorithms:

- **Algorithm A**: Use the $n_1$ samples from $p_t$ and estimate $p_t$ using denoising diffusion training.

- **Algorithm B**: Use $(n_1 + n_2)$ samples from the mixutre density $\tilde{p}_t = \frac{n_1}{n_1+n_2}p_t + \frac{n_2}{n_1+n_2}q_t$ and estimate $p_t$ using denoising diffusion training by pretending that all training samples are from $p_t$.

Using a connection between diffusion training and kernel density estimation, Daras et al. (2025c) propose a criterion for when to use Algorithm B over Algorithm A. Specifically, they show that, with probability $1 - \delta$ over the randomness in the samples, the error in total variation distance between the density estimated by Algorithm B and $p_t$ can be upper bounded by:

$$\text{Error}_{\text{using } n_1+n_2 \text{ samples from } \tilde{p}_t} \lesssim$$
$$\frac{1}{(n_1 + n_2)} + \frac{1}{\sigma_t^2(n_1 + n_2)} + d_{\text{TV}}(p_t, \tilde{p}_t) +$$
$$\sqrt{\frac{\log(n_1 + n_2) + \log(1 \vee \frac{n_1}{n_1+n_2}\lambda_1 + \frac{n_2}{n_1+n_2}\lambda_2) + \log 2/\delta}{\sigma_t^2(n_1 + n_2)}}, \tag{3}$$

while the error made by Algorithm A can be upper bounded by:

$$\text{Error}_{\text{using } n_1 \text{ samples from } p_t} \lesssim$$
$$\frac{1}{n_1} + \frac{1}{\sigma_t^2 n_1} + \sqrt{\frac{\log n_1 + \log(1 \vee \lambda_1) + \log 2/\delta}{\sigma_t^2 n_1}}, \tag{4}$$

where $\lesssim$ hides absolute constants in both cases. Thus, Daras et al. (2025c) propose a concrete criterion for when to choose Algorithm B over Algorithm A, namely for times $t$ such that:

$$\text{RHS of } equation\ 3 \leq \text{RHS of } equation\ 4. \tag{5}$$

For the purposes of applying the criterion, they also show that:

$$d_{\text{TV}}(p_t, \tilde{p}_t) \lesssim \frac{n_2}{\sigma_t(n_1 + n_2)} d_{\text{TV}}(p_0, q_0). $$

**Improved results through looping.** The theory of Ambient Omni compared (1) using only samples from the true distribution $p_t$ (Algorithm A), or (2) using samples from $\tilde{p}_t$

which is a mixture of the true distribution $p_t$ and the biased distribution $q_t$ (Algorithm B). However, there are more options. Our datalooping algorithm motivates the following alternate algorithm:

- **Algorithm C**: Transform samples from $q_t$ using a (potentially stochastic and learned) mapping function $f$. This defines the push-forward measure $\bar{q}_t = f\sharp q_t$. Then, learn using $(n_1 + n_2)$ samples from the distribution: $\tilde{\tilde{p}}_t = \frac{n_1}{n_1+n_2}p_t + \frac{n_2}{n_1+n_2}\bar{q}_t$.

Notice that Algorithm C is a generalization of Algorithm B, as the latter is recovered using the identity transformation function. Denote by $p_{t,\text{approx}}^{(L)}$ the approximate density estimated by Algorithm L, for $L \in \{A, B, C\}$. It is straightforward to show the following lemma:

**Lemma 4.1** (Contractive transformations lead to better learning). *If the mapping function $f$ contracts the TV distance with respect to the underlying true density $p_t$, i.e., if for any density $\phi$ it holds that:*

$$d_{\text{TV}}(f\sharp\phi, p_t) \leq d_{\text{TV}}(\phi, p_t), \tag{6}$$

*then, in all cases where Algorithm B is preferable to Algorithm A, according to Criterion 5, Algorithm C is weakly preferable to Algorithm B, and it is strictly preferable if equation 6 is strict.*

The lemma's statement is intuitive; if we have a way to "correct" the samples from the out-of-distribution density $q_t$, we should be able to achieve a better approximation to $p_t$ if we were to correct them versus using them as is. See appendix B.1 for the full proof. A related work (Gillman et al., 2024) studies the implications of having an idealized corrector function for learning from bad data (in their case, synthetic data) and establishes asymptotic convergence to the underlying distribution. Our result is similar in spirit, but the analysis is based on the bounds established by thinking about the implicit kernel-density estimation that diffusion modeling obtains.

With the above observations in place, let us identify conditions under which the approach of dataset refinement prescribed by Algorithm 1 would produce a correcting function $f$ that would allow Algorithm C to reduce the estimation error compared to the Algorithms A and B. In particular, suppose that the true scores $s_\tau(x) := \nabla_x \log p_\tau(x)$ of the distributions $(p_\tau)_\tau$ are only known approximately namely suppose that we have some estimated score function $\hat{s}_\tau(x) = \nabla_x \log p_\tau(x) + \varepsilon_\tau(x)$, and we use the estimated scores to perform posterior sampling of $X_t$ given $X_{t'}$, for $X_{t'} \sim q_{t'}$ at some $t' > t$. This would correspond to running the reverse diffusion process (Anderson, 1982; Oksendal, 2013) initializing at $X_{t'} \sim q_{t'}$ from time $t'$ down to time $t$:

$$dX_\tau = -\hat{s}_\tau(x)d\tau + d\bar{B}_\tau, \tag{7}$$

where $\bar{B}_t$ is a reverse time Brownian motion (that is 0 at time $t'$). Suppose that $f_{t',t}$ is the randomized map that takes a sample from $q_t$ adds Gaussian noise to it to produce a sample $X_{t'} \sim q_{t'}$, then runs the backwards diffusion equation 7 from $t'$ down to $t$. We show that under appropriate assumptions, the sampled distribution $f_{t',t} \# q_t$ is closer to $p_t$ compared to $q_t$. Thus, Algorithm C, using samples from $f_{t',t} \# q_t$ would have better estimation error compared to Algorithm B using samples from $q_t$ per Lemma 4.1.[1] We show the following for general dimensional measures. To state the theorem, suppose that $\rho_t$ is the measure of $X_t$ when we run the backwards diffusion process starting at $t' > t$, initializing $X_{t'} \sim q_{t'}$:

**Lemma 4.2** (Contraction of KL). *Let $f_{t',t}$ be defined as defined above. Suppose also that $p_0$ is supported in $B(0, R)$ and $t$ is large enough in $R$ or $p_0$ satisfies a log-Sobolev inequality with constant $C$. Then:*

$$D_{\mathrm{KL}}(f_{t',t} \# q_t, p_t) \le e^{-C_1(t'-t)} D_{\mathrm{KL}}(q_t, p_t) + C_2 \varepsilon,$$

*for some $C_1, C_2 < 1$ that depend on $C$ or $R$ (whichever is applicable) but not the dimension, and some $\varepsilon$ such that $\mathbb{E}_{\rho_\tau}[\|\varepsilon_\tau(X)\|^2] \le \varepsilon$ for all $\tau \in [t, t']$. See appendix B.2 for the proof.*

**Learning Errors.** Our looping framework *approximates* the score function with the *best estimator using the current data*. In particular, for times $t$ for which Algorithm B is preferred to Algorithm A, the estimation we have from the first round is:

$$\nabla \log p_{t,\mathrm{approx}}^{(2)}(x) =$$
$$\frac{1}{(n_1 + n_2)\sqrt{2\pi\sigma_t^2}} \left( \sum_{i=1}^{n_1} w(x, x_i)(x - x_i) + \sum_{i=1}^{n_2} w(x, x_i')(x - x_i) \right), \quad (8)$$

where $w(x, y) = \mathcal{N}(x; \mu = y, \sigma = \sigma_t^2)$, and $\{x_i\}_i$ are the samples from $p_t$ while $\{x_i'\}_i$ are the samples from $q_t$. This score only approximates the desired one, $\nabla \log p_t$. In practice, our experiments show that our estimates are sufficiently good estimates of the score function, and hence Algorithm C obtains faster rates of convergence than Algorithms A or B.

## 5. Experimental Results

**Experimental Setting.** We start our experiments by validating our approach in controlled settings. We follow the experimental methodology of the Ambient Omni paper; in particular, we train models on CIFAR-10 by corrupting $90\%$ of the dataset with Gaussian Blur and JPEG compression at various degradation levels while keeping $10\%$ of the dataset intact. We use the parameter $\sigma_B$ to refer to the standard deviation of the Gaussian kernel used for blurring the dataset

---

[1]Formally, to combine the second part of Lemma 4.2 with Lemma 4.1, one needs an adapation of the latter for KL.

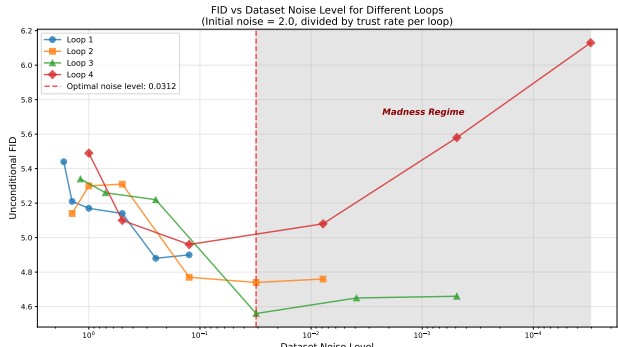

*Figure 3.* Multiple loops and ablation on the rate of progress. The horizontal axis is the noise level we denoise a corrupted CIFAR-10 dataset after $k$ loops, where $k$ changes for each one of the lines. Going too fast or too slow is suboptimal. There is a point after which reducing the dataset further only hurts (madness regime) because the current model has reached its denoising capacity. FID is always computed wrt to the original clean CIFAR-10.

images and the parameter $q$ to denote the file size after JPEG compression compared to the original file size.

We compare with the following baselines: **a)** quality-filtering (training only on the clean data), **b)** treating all-data as equal, and, **c)** Ambient Omni (Daras et al., 2025c), which is currently the state-of-the-art for learning diffusion generative models from corrupted data with unknown degradation types. We always initialize our method with the Ambient Omni checkpoints (loop 0). We further directly take the mapping between the low-quality samples (e.g. blurry/JPEG images) and their corresponding noising time (see Section 2 and F) from the work of Daras et al. (2025c), when needed. For all the experiments in this paper, we provide full details in Appendix Section E.

**Unconditional and Conditional Metrics.** We present unconditional FID results for all the baselines and one loop of our method in Table 1 (top). As shown, even a single loop of our proposed method leads to consistent and significant FID improvements up to $17\%$ reduction in FID. We emphasize that for all of the reported results, we always train the underlying models until performance saturates, and we report from the available checkpoints the one that achieves the best FID. This is to ensure that our method (which requires additional compute) does not have an unfair advantage. In the experimental settings we study, we are limited by the availability of data, not compute.

One benefit of starting our experimental analysis on this controlled setting is that we have the ground truth for the corrupted samples, and hence we can report conditional metrics too. We report conditional FID, LPIPS and MSE. Conditional FID is defined as follows; for each sample $(x_{t_i}, t_i)$ in the dataset we use a given model $h_\theta$ to sample from $\bar{X}_0 \sim p_{\theta,0}(\cdot | x_{t_i}, t_i)$, where $p_{\theta,0}(\cdot | x_{t_i}, t_i)$ is the distribution that arises by running the learned reverse process

*Table 1.* Unconditional and conditional results for CIFAR-10 with 90% corrupted and 10% clean data.

| Corruption | | Filtering | No Filtering | Ambient Omni (Loop 0) | | Ambient Dataloops (Loop 1) | |
|---|---|---|---|---|---|---|---|
| | | | | $\sigma_{\min}$ | FID ($\downarrow$) | $\sigma_{\min}$ | FID ($\downarrow$) |
| Blur | $\sigma_B = 0.6$ | 8.79 | 11.26 | $\sigma = 1.2$ | $5.689 \pm 0.0209$ | $\rho = 1.2/2^3$ | $4.947 \pm 0.0572$ |
| | $\sigma_B = 0.8$ | 8.79 | 28.26 | $\sigma = 1.9$ | $5.938 \pm 0.0583$ | $\rho = 1.9/2^3$ | $5.044 \pm 0.0709$ |
| | $\sigma_B = 1.0$ | 8.79 | 45.32 | $\sigma = 2.4$ | $6.080 \pm 0.0758$ | $\rho = 2.4/2^3$ | $5.358 \pm 0.0644$ |
| JPEG | $q = 50$ | 8.79 | 61.67 | $\sigma = 1.2$ | $5.836 \pm 0.0674$ | $\rho = 1.2/2^3$ | $4.825 \pm 0.0665$ |
| | $q = 25$ | 8.79 | 91.55 | $\sigma = 1.3$ | $6.188 \pm 0.0456$ | $\rho = 1.3/2^3$ | $5.531 \pm 0.0742$ |
| | $q = 18$ | 8.79 | 112.43 | $\sigma = 1.6$ | $6.261 \pm 0.0624$ | $\rho = 1.6/2^3$ | $5.464 \pm 0.0586$ |

**(a)** Unconditional metrics

| Corruption | | Dataset after Loop 0 | | | Dataset after Loop 1 | | |
|---|---|---|---|---|---|---|---|
| | | LPIPS $\downarrow$ | MSE $\downarrow$ | C-FID $\downarrow$ | LPIPS $\downarrow$ | MSE $\downarrow$ | C-FID $\downarrow$ |
| Blur | $\sigma_B = 0.6$ | 0.053 | 0.66 | $4.472 \pm 0.0694$ | 0.053 | 0.66 | $4.273 \pm 0.0394$ |
| | $\sigma_B = 0.8$ | 0.077 | 0.85 | $4.836 \pm 0.0377$ | 0.077 | 0.84 | $4.444 \pm 0.0221$ |
| | $\sigma_B = 1.0$ | 0.091 | 0.95 | $5.131 \pm 0.0244$ | 0.090 | 0.95 | $4.667 \pm 0.0074$ |
| JPEG | $q = 50$ | 0.052 | 0.67 | $4.412 \pm 0.0181$ | 0.051 | 0.67 | $4.142 \pm 0.0367$ |
| | $q = 25$ | 0.058 | 0.72 | $4.942 \pm 0.0485$ | 0.058 | 0.72 | $5.043 \pm 0.0291$ |
| | $q = 18$ | 0.070 | 0.80 | $5.014 \pm 0.0103$ | 0.069 | 0.80 | $4.935 \pm 0.0083$ |

**(b)** Conditional metrics

initialized at time $t = t_i$ with the noisy sample $x_{t_i}$. We then compute the FID between the set sampled with posterior sampling and the reference set. MSE and LPIPS are point-wise restoration metrics and hence it is more meaningful to compute them by measuring the distance of the ground-truth sample to the posterior mean, rather than any random sample from the posterior distribution. In particular, for each sample $(x_{t_i}, t_i)$ in the dataset we use a given model $h_\theta$ to estimate with Monte Carlo the posterior mean defined as $\mathbb{E}_{\bar{X}_0 \sim p_{\theta,0}(\cdot | x_{t_i}, t_i)}[\bar{X}_0]$. For a perfectly trained model and ignoring discretization errors, this quantity equals $h_\theta(x_{t_i}, t_i)$, but we use the former quantity to account for learning and sampling errors. We report our conditional results in Table 1 (bottom). Interestingly, although unconditional FID is always better for the model after the loop, this is not always the case for the conditional metrics. A corollary is that if we use the L1 model to restore the dataset, we might yield worse performance compared to stopping after 1 loop. This can happen, as seen below.

**Multiple loops and rate of progress.** Roughly speaking, there are two reasons that can lead to deterioration in performance. The first has to do with the inherent limit on how much a finite dataset can be denoised reliably. Attempting to go beyond this limit will cause any algorithm to fail. The second reason has to do with the optimization (looping) process, i.e. with *how we reach the denoising limit*. We investigate this extensively in Figure 3 and Table 7. The figure shows the effect of bringing the dataset to a noise level by running one or more loops. Going too fast or too slow is hurtful, due to overconfidence in the model's capabilities or accumulation of errors, respectively. We provide guidance on how to select the denoising rate in Appendix Section C.2. Further, there is a limit to how much we can safely denoise the dataset, after which limit we reach the "madness regime" (Alemohammad et al., 2024a) where performance degrades significantly. The optimal performance for the blurring corruption with $\sigma_B = 0.6$ is achieved with

3 loops, where the dataset noise level is reduced by 8 each time. On the other hand, if we cannot afford running multiple loops, Table 7 suggests that taking a larger denoising step for one loop is preferable.

**Other ablations.** Beyond the number of loops and the rate of denoising progress, we provide numerous ablations in the Appendix that quantify the role of different aspects of our approach. In particular, Figure 6 shows that the improvements are across all diffusion times, Table 10 shows that there are benefits in sampling multiple times from the posterior, and 9 shows the effect of restoring the dataset of the previous round compared to always restoring the original dataset. The main takeaway is that by carefully tuning parts of the pipeline, we can further boost performance. For example, in the Appendix, we manage to push the unconditional FID for $\sigma_B = 0.6$ on CIFAR from the 5.34 reported in Omni all the way to 4.52. While such improvements are possible, we run the majority of the experiments in the main paper with the simplest variant of our method, as it achieves comparable performance and is far more educational to the reader.

## 5.1. Experiments with synthetic data and text-to-image models

Having established the effectiveness of the method in controlled settings, we are now ready to test our algorithm in real use cases. In particular, we experiment with text-to-image generative modeling, following the architectural and dataset choices of MicroDiffusion (Sehwag et al., 2025). Sehwag et al. (2025) train a diffusion model from scratch using only 8 GPUs in 2 days. During that training, 4 datasets are used; Conceptual Captions (12M) (Sharma et al., 2018), Segment Anything (11M) (Kirillov et al., 2023), JourneyDB (4.2M) (Sun et al., 2023), and DiffusionDB (10.7M) (Wang et al., 2022). Daras et al. (2025c) noticed that DiffusionDB, despite contributing 28.23% of the dataset samples, contains synthetic images that have significantly lower quality than the rest of the dataset. To account for this, the authors noise the DiffusionDB dataset to level $\sigma_{\text{DiffusionDB}} = 2.0$ and only use it to train for diffusion times $t : \sigma_t \geq 2.0$. This leads to a significant COCO FID improvement compared to using it as clean; FID drops from 12.37 to 10.61.

We now attempt to further improve the performance by taking the model trained by Daras et al. (2025c) to denoise the DiffDB dataset and then train a new model on the denoised set. Consistent with the description of our algorithm in Section 3, we do partial dataset restoration by performing posterior sampling to bring the DiffDB dataset at noise level $\sigma'_{\text{DiffusionDB}} = \sigma_{\text{DiffusionDB}}/2 = 1.0$. We then train the model on this denoised dataset, using the Ambient Diffusion training objective equation 2, as usual. The resulting model achieves further improvements to COCO FID and CLIP-

*Table 2.* Quantitative results on COCO zero-shot generation.

| Method | FID-30K ($\downarrow$) | Clip-FD-30K ($\downarrow$) |
|---|---|---|
| Micro-diffusion | 12.37 | 10.07 |
| Ambient-o (L0) | 10.61 | 9.40 |
| Ambient Loops (L1) | **10.06** | **8.83** |

*Table 3.* Quantitative results on GenEval (Ghosh et al., 2023).

| Method | Overall | Single | Two | Counting | Colors | Position | Color attr. |
|---|---|---|---|---|---|---|---|
| Micro-diffusion | 0.44 | 0.97 | 0.33 | 0.35 | 0.82 | 0.06 | 0.14 |
| Ambient-o (L0) | 0.47 | 0.97 | **0.40** | **0.36** | 0.82 | 0.11 | 0.14 |
| Ambient Loops (L1) | **0.47** | **0.97** | 0.38 | 0.35 | 0.78 | **0.11** | **0.19** |

FD score, as shown in Table 2, and comparable GenEval scores as shown in Table 3. The fact that we get comparable GenEval scores is expected, as this benchmark measures text-to-image alignment, and our method does not explicitly target corruptions in the text-conditioning signal. Figure 1 shows examples of images from DiffDB and their evolution across our looping process. As seen, the datasets seem to be converging after 1 loop.

**Comparisons with inference-time guidance methods.** Our Dataloops approach addresses imperfections in the dataset at training-time by co-evolving the model and the imperfect dataset. Another possibility is to correct the distribution of a model trained on imperfect data at *test-time*. One such approach is proposed in SIMS (Alemohammad et al., 2024b). The authors use a guidance method to steer the distribution towards the best-quality part of the dataset and away from synthetic outputs produced by another model. Inspired by this approach, we train two models: (a) a model trained on all the datasets, including the lower-quality synthetic DiffDB dataset, and (b) a model trained without the low-quality synthetic DiffDB dataset. We use SIMS to guide towards (b) and away from (a). The results in terms of quality and COCO FID are shown in Table 4. As shown, increasing the guidance improves the quality of the generated images but at the expense of diversity, which leads to overall worse FID for high guidance strengths.

| Method | COCO FID | CLIP-IQA |
|---|---|---|
| SIMS - guidance $\omega = 0.0$ | 13.03 | 0.4171 |
| SIMS - guidance $\omega = 0.05$ | 13.05 | 0.4204 |
| SIMS - guidance $\omega = 0.1$ | 13.04 | 0.4210 |
| SIMS - guidance $\omega = 0.25$ | 13.08 | 0.4207 |
| SIMS - guidance $\omega = 0.5$ | 13.23 | **0.4213** |
| Ambient Loops (L1) | **8.83** | 0.4171 |

*Table 4.* Comparisons with SIMS (Alemohammad et al., 2024b) on COCO FID and CLIP-IQA for different guidance strengths.

## 5.2. ImageNet Experiments

To show the universality of our approach, for our next experiment, we then set out to improve the quality of certain images from the ImageNet dataset. We use a quality classifier to separate the bottom 20% of ImageNet in terms of quality, and then we use the ImageNet generative model trained in (Degeorge et al., 2025) to improve it. Table 5 shows the quality improvements of the resulting dataset across different reward models. Finetuning the model from (Degeorge et al., 2025) on this dataset leads to improved generations according to 3/4 reward metrics as shown in Table 6.

| Metric | Original ImageNet | "Restored" ImageNet |
|---|---|---|
| Aesthetic | $4.03 \pm 0.34$ | $\mathbf{4.41} \pm 0.48$ |
| HPSv2 | $0.20 \pm 0.03$ | $\mathbf{0.23} \pm 0.03$ |
| ImgReward | $-0.23 \pm 0.88$ | $\mathbf{0.24} \pm 0.89$ |
| PickScore | $20.39 \pm 1.25$ | $\mathbf{20.70} \pm 1.21$ |

*Table 5.* Quality improvements on ImageNet by using the generative model of Degeorge et al. (2025) to improve the quality of the bottom 20% of images.

| Model | PickScore | Aes. Score | HPSv2 | ImageReward |
|---|---|---|---|---|
| Base | 20.07 | **4.85** | 0.22 | -0.37 |
| Loops | **20.13** | 4.81 | **0.24** | **-0.19** |

*Table 6.* Finetuning results on an improved version of ImageNet that we get by refining the bottom 20% of the original dataset with outputs of the ImageNet generative model of Degeorge et al. (2025).

## 5.3. De-novo protein design

**Introduction.** For this final part of our experimental evaluation, we switch modality and target structural protein design. This problem is significant because accurate de novo protein structure models can result in improved designs for new vaccines, therapeutics, and enzymes. The problem is also well-suited for our Ambient Dataloops framework, as techniques for determining the atomistic resolution of molecular protein structures (such as X-ray crystallography) are inherently noisy. On top of that, acquiring samples through such techniques requires domain expertise and significant resources and hence the available datasets, such as the Protein Data Bank, are of limited size. To enrich the dataset, recent state-of-the-art models for protein backbones are trained on synthetic data from AlphaFold, which once again contain corrupted samples due to learning errors.

**Setting.** Daras et al. (2025b) applied the Ambient Omni (Daras et al., 2025c) framework to train a generative model for protein backbones. We use the same dataset, architectural, and training procedures as in (Daras et al., 2025b) to demonstrate that looping can improve perfor-

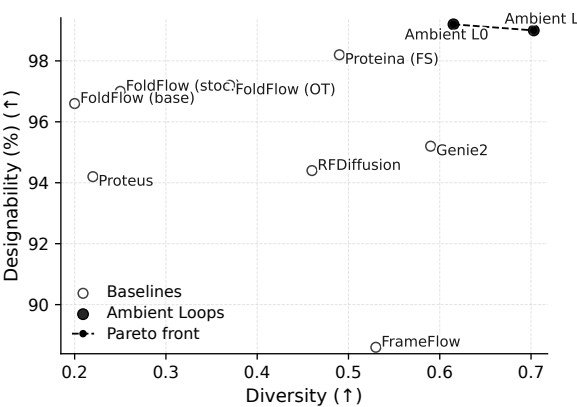

*Figure 4.* Designability-Diversity trade-off for de novo design of protein backbones. Training with Ambient Proteins dominates the Pareto frontier. One loop of our framework achieves a 14.3% increase in diversity for a minor 0.2% in designability.

mance in domains beyond Computer Vision. In particular, we start with the dataset of Daras et al. (2025b) that contains 90,250 structurally unique proteins from the AlphaFold Data Bank (AFDB) dataset, with a maximum length of 256 residues. To find the associated noise level of the dataset we follow once again the experimental protocol of the authors, which is to map proteins to diffusion times according to AlphaFold's self-reported confidence for the predicted structure as given by the pLDDT score. We then use the Ambient Proteins (Daras et al., 2025b) model to denoise its training set and we start a new training run on the denoised dataset. One example of such a denoising is given in Figure 5. In agreement with the rest of the paper, we also treat the denoised dataset as noisy, but at a lower noise level. In this particular domain, we use the existing pLDDT to diffusion time mapping from Ambient Proteins (Daras et al., 2025b) and we treat the denoised predictions as increasing the pLDDT (synthetically) by 3 points in each denoised sample. We arrived at this value after ablating different pLDDT adjustments that led to inferior results. To assess the quality of the trained models, we use the two most established metrics in the field: Designability and Diversity. There is a trade-off between the two metrics that defines a Pareto frontier in the joint space.

**Results.** Just one loop of our procedure is enough to achieve a new Pareto point, as shown in Figure 4. In particular, we trade 0.2% decrease in designability for a 14.3% increase in diversity, significantly expanding the creativity boundaries of the loop 0 model for the same inference parameters. Both models dominate in the Pareto frontier over other baselines showing both the promise of degradation-aware diffusion training and the potential of datalooping to enhance the generative capabilities for protein design. While our protein evaluation is preliminary and the results need to be verified in the wet lab, the metrics suggest that datalooping could be useful for scientific domains.

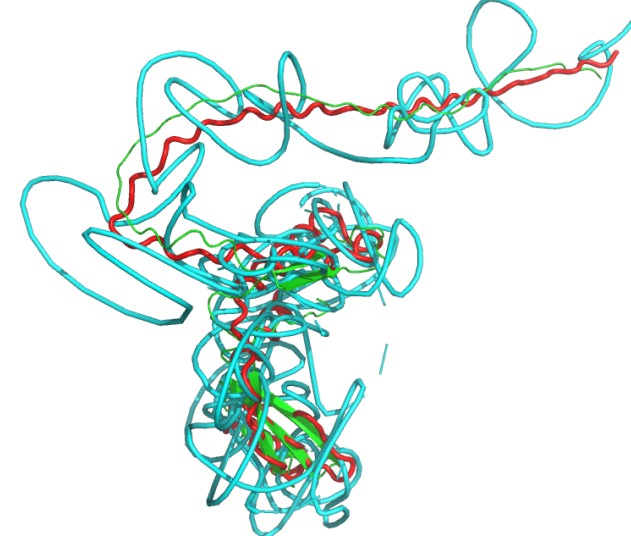

*Figure 5.* Example of our dataset refinement procedure. An initial low pLDDT protein, denoted with green, is noised to a certain level, giving the shape in cyan. We initialize the reverse process with the cyan sample, and we sample the red point from the posterior.

## 6. Conclusions and Future Work

We introduced Ambient Dataloops, a framework that enables better learning of the underlying data distribution by refining the dataset together with the model being trained. This algorithm paves the way for denoising scientific datasets where sample quality naturally varies and it has the potential to improve not only generative models but also supervised models optimized for downstream applications.

## Acknowledgements

This research is supported by a Simons Investigator Award, a Simons Collaboration on Algorithmic Fairness, ONR MURI grant N00014-25-1-2116, ONR grant N00014-25-1-2296 and ONR MURI grant N033697-00007. The experiments were run on the Vista GPU Cluster through the Center for Generative AI (CGAI) and the Texas Advanced Computing Center (TACC) at UT Austin. Adrian Rodriguez-Munoz is supported by a DSTA, Singapore grant.

## Impact Statement

This paper presents work whose goal is to advance the field of Machine Learning, and thus we have published our code here (https://github.com/adrianrm99/ambient_dataloops) for the community. Given that (1) all the datasets we used are in the public domain and (2) prior works have already made public models trained on this data, we do not believe our work introduces extra risks that do not already exist.

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

# A. Ambient Dataloops Training Algorithm

---

**Algorithm 1** Ambient Dataloops Training Algorithm.

---

**Require:** dataset $\mathcal{D}^{(0)} = \{(x_{t_i}, t_i)\}_{i=1}^{N}$, noise scheduling $\sigma(t)$, batch size $B$, diffusion time $T$, number of loops $L$, random weights $\theta^{(0)}$.

 0: **for** $l \in [1, L]$ **do** {A new loop starts.}

 0:     $\theta^{(l)} \leftarrow \theta^{(l-1)}$ {Initialize from the weights of the previous round (finetuning).}

 0:     **while** not converged **do** {A new training starts.}

 0:         $t_{s_1}, ..., t_{s_B} \leftarrow$ Sample uniformly B times in $[0, T]$

 0:         $(x_{\bar{t}_1}, \bar{t}_1), ..., (x_{\bar{t}_B}, \bar{t}_B) \leftarrow$ Sample (at random) points from $\mathcal{D}^{(l-1)}$ that can be used for times $t_{s_1}, ..., t_{s_B}$, i.e.

    points that have lower noise level than the corresponding $t_{s_i}$.

 0:         loss $\leftarrow 0$ {Initialize loss.}

 0:         **for** $(x_{\bar{t}_i}, \bar{t}_i, t_{s_i}) \in (x_{\bar{t}_1}, \bar{t}_1, t_{s_1}), ..., (x_{\bar{t}_B}, \bar{t}_B, t_{s_B})$ **do**

 0:             $\epsilon \sim \mathcal{N}(0, I)$ {Sample noise.}

 0:             $x_{t_{s_i}} \leftarrow x_{\bar{t}_i} + \sqrt{\sigma^2(t_{s_i}) - \sigma^2(\bar{t}_i)}\epsilon$ {Add additional noise.}

 0:             $\alpha(t_{s_i}, \bar{t}_i) \leftarrow \frac{\sigma^2(t_{s_i}) - \sigma^2(\bar{t}_i)}{\sigma^2(t_{s_i})},$

 0:             $w(t_{s_i}, \bar{t}_i) \leftarrow \frac{\sigma^4(t_{s_i})}{\left(\sigma^2(t_{s_i}) - \sigma^2(\bar{t}_i)\right)^2}$. {Loss reweighting.}

 0:             loss $\leftarrow$ loss $+ w(t_{s_i}, \bar{t}_i) \left\| \alpha(t_{s_i}, \bar{t}_i) h_{\theta^{(l)}}(x_{t_{s_i}}, t_{s_i}) + (1 - \alpha(t_{s_i}, \bar{t}_i))x_{t_{s_i}} - x_{\bar{t}_i} \right\|^2$

 0:         **end for**

 0:         loss $\leftarrow \frac{\text{loss}}{B}$ {Compute average loss.}

 0:         $\theta^{(l)} \leftarrow \theta^{(l)} - \eta\nabla_{\theta^{(l)}}$ loss {Update network parameters via backpropagation.}

 0:     **end while**

 0:     $\mathcal{D}^{(l)} = \emptyset$

 0:     **for** $(x_{t_i}, t_i) \in \mathcal{D}^{(0)}$ **do** {A new restoration cycle starts.}

 0:         $x_{t_i/2^l} \sim p_{\theta^{(l)}, t_i/2^l}(\cdot|x_{t_i}, t_i)$ {Perform posterior sampling from $t_i$ to $t_i/2^l$.}

 0:         $\mathcal{D}^{(l)} \leftarrow \mathcal{D}^{(l)} \cup (x_{t_i/2^l}, t_i/2^l)$ {Add restored point to dataset.}

 0:     **end for**

 0: **end for**=0

---

# B. Theoretical Results from Section 4

**Lemma B.1** (Contractive transformations lead to better learning; Lemma 4.1 restated). *If the mapping function $f$ contracts the TV distance with respect to the underlying true density $p_t$, i.e., if for any density $\phi$ it holds that:*

$$d_{\text{TV}}(f\sharp\phi, p_t) \leq d_{\text{TV}}(\phi, p_t), \tag{9}$$

*then, in all cases where Algorithm B is preferable to Algorithm A, according to Criterion 5, Algorithm C is weakly preferable to Algorithm B, and it is strictly preferable if equation 9 is strict.*

*Proof of Lemma 4.1:* We bound the estimation error made by Algorithms B and C using equation 3. The bounds are the same except that the bound for the error of Algorithm B has an additive term $d_{\text{TV}}(p_t, \tilde{p}_t)$ while the bound for the error of Algorithm C replaces that term with $d_{\text{TV}}(p_t, \tilde{\tilde{p}}_t)$. By definition of $\tilde{p}_t$ and $\tilde{\tilde{p}}_t$, we have:

$$d_{\text{TV}}(p_t, \tilde{p}_t) = \frac{n_2}{n_1 + n_2} d_{\text{TV}}(p_t, q_t),$$

$$d_{\text{TV}}(p_t, \tilde{\tilde{p}}_t) = \frac{n_2}{n_1 + n_2} d_{\text{TV}}(p_t, \bar{q}_t) = \frac{n_2}{n_1 + n_2} d_{\text{TV}}(p_t, f\#q_t).$$

Thus, given equation 9:

$$d_{\text{TV}}(p_t, \tilde{\tilde{p}}_t) \leq d_{\text{TV}}(p_t, \tilde{p}_t),$$

and this inequality becomes strict if equation 9 is strict. Thus, Criterion 5 always weakly prefers Algorithm C to Algorithm B, and the preference is strict if equation 9 is strict. $\square$

**Lemma B.2** (Contraction of KL; Lemma 4.2 restated)**.** *Let $f_{t',t}$ be defined as in Section 4. Suppose also that $p_0$ is supported in $B(0, R)$ and $t$ is large enough with respect to $R$ or $p_0$ satisfies a log-Sobolev inequality with constant $C$. Then:*

$$D_{\mathrm{KL}}(f_{t',t}\#q_t, p_t) \leq e^{-C_1(t'-t)}D_{\mathrm{KL}}(q_t, p_t) + C_2\varepsilon,$$

*for some $C_1, C_2 < 1$ that depend on $C$ or $R$ (whichever is applicable) but not the dimension, and some $\varepsilon$ such that $\mathbb{E}_{\rho_\tau}[\|\varepsilon_\tau(X)\|^2] \leq \varepsilon$ for all $\tau \in [t, t']$.*

*Proof of Lemma 4.2:* Recall that a measure $\nu$ satisfies the log-Sobolev inequality (LSI) with a constant $C > 0$ if for all smooth functions $g$ with $\mathbb{E}_\nu[g^2] \leq \infty$ :

$$\mathbb{E}_\nu[g^2 \log g^2] - \mathbb{E}_\nu[g^2] \log \mathbb{E}_\nu[g^2] \leq 2C\mathbb{E}_\nu[\|\nabla g\|^2]. \tag{10}$$

It is known that if $p_0$ is supported on the ball $B(0, R)$, then $p_t = p_0 \circledast \mathcal{N}(0, \sigma_t^2 I)$ satisfies the log-Sobolev inequality with constant $6(4R^2 + \sigma_t^2)e^{\frac{4R^2}{\sigma_t^2}}$ (Chen et al., 2021). It is also known that if $p_0$ satisfies the log-Sobolev inequality with some constant $C$, then $p_t = p_0 \circledast \mathcal{N}(0, \sigma_t^2 I)$ satisfies the log-Sobolev inequality with constant $C + \sigma_t^2$ (Courtade & Wang, 2025). In both cases, the measures $p_t$ for $t$ bounded away from 0 satisfy the log-Sobolev inequality for some constant independent of the dimension. In the second case, this is true for all $t$. Let us call $C'$ the log-Sobolev constant satisfied by all $p_\tau, \tau \in [t, t']$.

Next, recall that we denote by $\rho_\tau(x)$ the distribution of $X_\tau$ when we run the backward SDE 7, initialized at $X_{t'} \sim q_{t'}$, for some $t' > t$. For convenience, we define the functions $\tilde{\rho}_\tau(x) = \rho_{t'-\tau}(x)$, $\tilde{p}_\tau(x) = p_{t'-\tau}(x)$ and $\tilde{\varepsilon}_\tau(x) = \varepsilon_{t'-\tau}(x)$, for all $\tau \in [0, t']$.

Recall that the distributions $(p_\tau)_\tau$ are the marginals of the forward SDE, which is a diffusion process $dX_t = dB_t$ where $B_t$ is a forward time Brownian motion. The Fokker-Planck equation provides us with the time derivative of $p_\tau$ and therefore $\tilde{p}_\tau$:

$$\frac{\partial p_\tau}{\partial \tau} = \frac{1}{2}\Delta_x p_\tau,$$

where $\Delta_x$ is the Laplacian operator. Thus:

$$\frac{\partial \tilde{p}_\tau}{\partial \tau} = -\frac{1}{2}\Delta_x \tilde{p}_\tau \tag{11}$$

On the other hand, the Fokker-Planck equation for the backward SDE 7 gives:

$$\begin{aligned}
\frac{\partial \tilde{\rho}_\tau}{\partial \tau} &= -\nabla_x \cdot (\tilde{\rho}_\tau \hat{s}_{t'-\tau}) + \frac{1}{2}\Delta_x \tilde{\rho}_\tau \\
&= -\nabla_x \cdot (\tilde{\rho}_\tau(\nabla_x \log \tilde{p}_\tau + \tilde{\varepsilon}_\tau)) + \frac{1}{2}\Delta_x \tilde{\rho}_\tau \\
&= \nabla_x \cdot \left(-\frac{1}{2}\tilde{\rho}_\tau \nabla_x \log \tilde{p}_\tau - \tilde{\rho}_\tau \cdot \tilde{\varepsilon}_\tau + \frac{1}{2}\tilde{\rho}_\tau \nabla_x \log \frac{\tilde{\rho}_\tau}{\tilde{p}_\tau}\right)
\end{aligned} \tag{12}$$

Now, for two measures $\mu, \nu$ on $\mathbb{R}^n$ defined by their probability density functions $\mu(x), \nu(x)$ let us denote by

$$H_\nu(\mu) = \mathrm{KL}(\mu\|\nu) \equiv \int_{\mathbb{R}^n} \mu(x) \log \frac{\mu(x)}{\nu(x)}dx,$$

the KL divergence of $\mu$ with respect to $\nu$. And let us denote by

$$J_\nu(\mu) = \int_{\mathbb{R}^n} \mu(x)\left\|\nabla \log \frac{\mu(x)}{\nu(x)}\right\|^2 dx,$$

the relative Fisher information of $\mu$ with respect to $\nu$.

The remainder will generalize the derivation of (Vempala & Wibisono, 2019) for Langevin diffusion to our non-stationary SDE 7. Via a straightforward calculation we have the following:

$$\frac{d}{d\tau}H_{\tilde{p}_\tau}(\tilde{\rho}_\tau) = \int \log\left(\frac{\tilde{\rho}_\tau}{\tilde{p}_\tau}\right)\frac{\partial}{\partial \tau}\tilde{\rho}_\tau dx - \int \frac{\tilde{\rho}_\tau}{\tilde{p}_\tau}\frac{\partial}{\partial \tau}\tilde{p}_\tau \, dx. \tag{13}$$

Plugging equation 12 into the first term of equation 13 and using integration by parts, we have:

$$\int \log \left( \frac{\tilde{\rho}_\tau}{\tilde{p}_\tau} \right) \frac{\partial}{\partial \tau} \tilde{\rho}_\tau dx = \int \log \left( \frac{\tilde{\rho}_\tau}{\tilde{p}_\tau} \right) \nabla_x \cdot \left( -\frac{1}{2} \tilde{\rho}_\tau \nabla_x \log \tilde{p}_\tau - \tilde{\rho}_\tau \cdot \tilde{\varepsilon}_\tau + \frac{1}{2} \tilde{\rho}_\tau \nabla_x \log \frac{\tilde{\rho}_\tau}{\tilde{p}_\tau} \right) dx$$

$$= -\int \nabla_x \log \left( \frac{\tilde{\rho}_\tau}{\tilde{p}_\tau} \right) \cdot \left( -\frac{1}{2} \tilde{\rho}_\tau \nabla_x \log \tilde{p}_\tau - \tilde{\rho}_\tau \cdot \tilde{\varepsilon}_\tau + \frac{1}{2} \tilde{\rho}_\tau \nabla_x \log \frac{\tilde{\rho}_\tau}{\tilde{p}_\tau} \right) dx$$

$$= -\frac{1}{2} \int \tilde{\rho}_\tau \left\| \nabla_x \log \left( \frac{\tilde{\rho}_\tau}{\tilde{p}_\tau} \right) \right\|^2 dx + \int \nabla_x \log \left( \frac{\tilde{\rho}_\tau}{\tilde{p}_\tau} \right) \cdot \left( \frac{1}{2} \tilde{\rho}_\tau \nabla_x \log \tilde{p}_\tau + \tilde{\rho}_\tau \cdot \tilde{\varepsilon}_\tau \right) dx \quad (14)$$

Plugging equation 11 into the second term of equation 13 and using integration by parts, we have:

$$\int \frac{\tilde{\rho}_\tau}{\tilde{p}_\tau} \frac{\partial}{\partial \tau} \tilde{p}_\tau \, dx = -\frac{1}{2} \int \frac{\tilde{\rho}_\tau}{\tilde{p}_\tau} \Delta_x \tilde{p}_\tau \, dx$$

$$= -\frac{1}{2} \int \frac{\tilde{\rho}_\tau}{\tilde{p}_\tau} \nabla_x \cdot \nabla_x \tilde{p}_\tau \, dx$$

$$= \frac{1}{2} \int \nabla_x \frac{\tilde{\rho}_\tau}{\tilde{p}_\tau} \cdot \nabla_x \tilde{p}_\tau \, dx$$

$$= \frac{1}{2} \int \frac{\tilde{\rho}_\tau}{\tilde{p}_\tau} \nabla_x \log \frac{\tilde{\rho}_\tau}{\tilde{p}_\tau} \cdot \nabla_x \tilde{p}_\tau \, dx$$

$$= \frac{1}{2} \int \tilde{\rho}_\tau \nabla_x \left( \log \frac{\tilde{\rho}_\tau}{\tilde{p}_\tau} \right) \cdot \nabla_x \log \tilde{p}_\tau \, dx \quad (15)$$

Combining equation 13, equation 14 and equation 15 we have:

$$\frac{d}{d\tau} H_{\tilde{p}_\tau}(\tilde{\rho}_\tau) = -\frac{1}{2} \int \tilde{\rho}_\tau \left\| \nabla_x \log \left( \frac{\tilde{\rho}_\tau}{\tilde{p}_\tau} \right) \right\|^2 dx + \int \tilde{\rho}_\tau \tilde{\varepsilon}_\tau \nabla_x \log \left( \frac{\tilde{\rho}_\tau}{\tilde{p}_\tau} \right) dx$$

By the Cauchy-Schwarz inequality, we have that $u \cdot v \leq \alpha ||u||^2 + \frac{1}{4\alpha} ||v||^2$ for all vectors $u, v$ and $\alpha > 0$. So we get that for all $\alpha \in (0, 1/2)$:

$$\frac{d}{d\tau} H_{\tilde{p}_\tau}(\tilde{\rho}_\tau) \leq -\frac{1}{2} \int \tilde{\rho}_\tau \left\| \nabla_x \log \left( \frac{\tilde{\rho}_\tau}{\tilde{p}_\tau} \right) \right\|^2 dx + \alpha \int \tilde{\rho}_\tau \left\| \nabla_x \log \left( \frac{\tilde{\rho}_\tau}{\tilde{p}_\tau} \right) \right\|^2 dx + \frac{1}{4\alpha} \int \tilde{\rho}_\tau ||\tilde{\varepsilon}_\tau||^2 dx$$

$$= -\left( \frac{1}{2} - \alpha \right) J_{\tilde{p}_\tau}(\tilde{\rho}_\tau) + \frac{1}{4\alpha} \int \tilde{\rho}_\tau ||\tilde{\varepsilon}_\tau||^2 dx \quad (16)$$

Note that, if $\nu$ satisfies the log-Sobolev inequality with constant $C'$, this immediately implies that for all measures $\mu$ (by setting $g^2 = \frac{\mu}{\nu}$ in equation 10):

$$H_\nu(\mu) \leq \frac{C'}{2} J_\nu(\mu).$$

Because all measures $p_\tau$ satisfy the log-Sobolev inequality with constant $C'$ for all $\tau \in [t, t']$, plugging this into equation 16:

$$\frac{d}{d\tau} H_{\tilde{p}_\tau}(\tilde{\rho}_\tau) \leq -\frac{1 - 2\alpha}{C'} H_{\tilde{p}_\tau}(\tilde{\rho}_\tau) + \frac{1}{4\alpha} \int \tilde{\rho}_\tau ||\tilde{\varepsilon}_\tau||^2 dx$$

$$= -\frac{1 - 2\alpha}{C'} H_{\tilde{p}_\tau}(\tilde{\rho}_\tau) + \frac{1}{4\alpha} \mathbb{E}_{\tilde{\rho}_\tau}[||\tilde{\varepsilon}_\tau||^2] \quad (17)$$

$$= -\frac{1 - 2\alpha}{C'} H_{\tilde{p}_\tau}(\tilde{\rho}_\tau) + \frac{\varepsilon}{4\alpha}. \quad (18)$$

Denoting by $y(\tau) = H_{\tilde{p}_\tau}(\tilde{\rho}_\tau)$, the above can be written as:

$$\frac{d}{d\tau} y(\tau) \leq -\frac{1 - 2\alpha}{C'} y(\tau) + \frac{\varepsilon}{4\alpha}. \quad (19)$$

Integrating, we have that for $\tau \in [0, t' - t]$:

$$y(\tau) \leq e^{-\frac{1-2\alpha}{C'}\tau} y(0) + \left( 1 - e^{-\frac{1-2\alpha}{C'}\tau} \right) \frac{\varepsilon C'}{4\alpha(1 - 2\alpha)}.$$

Recalling that $y(\tau) = H_{\tilde{p}_\tau}(\tilde{\rho}_\tau) = H_{p_{t'-\tau}}(\rho_{t'-\tau})$, we have that, for all $\tau \in [t, t']$:

$$H_{p_\tau}(\rho_\tau) \le e^{-\frac{1-2\alpha}{C'}(t'-\tau)} H_{p_{t'}}(\rho_{t'}) + \left(1 - e^{-\frac{1-2\alpha}{C'}(t'-\tau)}\right) \frac{\varepsilon C'}{4\alpha(1-2\alpha)},$$

which shows that $H_{p_\tau}(\rho_\tau)$ contracts exponentially fast as long as it is larger than $\frac{\varepsilon C'}{4\alpha(1-2\alpha)}$. $\square$

## C. Additional image results and ablations

### C.1. Multiple loops and rate of progress

We show results for different numbers of loops and denoising rates in Table 7.

*Table 7.* Ablation on number of Dataloops and rate of progress for CIFAR10-32x32 under blur corruption ($\sigma_B = 0.6$). Original $\sigma_{\min} = 1.2$. Metric: FID $\downarrow$.

| Trust rate $\rho$ (noise level) | Ambient Omni (Loop 0) | | Loop 1 | | Loop 2 | | Loop 3 | | Loop 4 | |
|---|---|---|---|---|---|---|---|---|---|---|
| | Uncond. FID | Cond. FID | Uncond. FID | Cond. FID | Uncond. FID | Cond. FID | Uncond. FID | Cond. FID | Uncond. FID | Cond. FID |
| $2^{0.25}$ | 5.67 | 4.46 | 5.44 | 4.46 | 5.14 | 4.14 | 5.34 | 4.16 | 5.49 | 4.18 |
| $2^{0.50}$ | 5.67 | 4.46 | 5.21 | 4.26 | 5.30 | 4.10 | 5.26 | 4.00 | 5.10 | 4.00 |
| $2^{1.00}$ | 5.67 | 4.46 | 5.17 | 4.21 | 5.31 | 4.08 | 5.22 | 3.91 | 4.96 | 3.61 |
| $2^{2.00}$ | 5.67 | 4.46 | 5.14 | 4.03 | 4.77 | **3.53** | 4.56 | 3.76 | 5.08 | 5.09 |
| $2^{3.00}$ | 5.67 | 4.46 | 4.88 | 3.98 | **4.74** | 3.72 | 4.65 | 4.25 | 5.58 | 5.68 |
| $2^{4.00}$ | 5.67 | 4.46 | 4.90 | 4.04 | 4.76 | 3.98 | 4.66 | 5.19 | 6.13 | 8.02 |
| $\infty$ | 5.67 | 4.46 | 5.03 | 4.14 | 4.99 | 4.86 | 5.86 | 7.92 | 8.93 | 12.69 |

### C.2. Predicting the best denoising rate

As shown in the previous section and Table 7, the rate at which we denoise the dataset is a very important hyperparameter that controls the success of the proposed algorithm. To avoid the expensive sweeping of this hyperparameter, we provide some guidance on how to select it. As it turns out, the conditional FID of the first loop is a highly predictive metric for the best noise level schedule for subsequent loops. We show this by computing the Spearman rank correlations for unconditional FID of the next loop with the conditional FIDs of the previous loop in Table 8.

| Transition | Uncond FID | Cond FID |
|---|---|---|
| Loop 1 → Loop 2 | 0.82 | 0.82 |
| Loop 2 → Loop 3 | 0.54 | 1.00 |
| Loop 3 → Loop 4 | 0.36 | 0.96 |
| Loop 1 → Best Uncond | 0.71 | 0.93 |

*Table 8.* Spearman rank correlations for unconditional FID of the next loop with the unconditional and conditional FIDs of the previous loop.

### C.3. Number of posterior samples

A natural ablation to consider is the multiplicity of posterior samplings performed during restoration. Concretely, while for all our experiments in the main paper we did posterior sampling exactly once for each corrupted sample, we can also choose to sample many times from the model using the same corrupted image. As this effectively multiplies the amount of corrupted data in our training set, we duplicate the clean data by the same amount to maintain balance. We see results for training on the multiplied datasets in Table 10 in the case of Blur ($\sigma_B = 0.6$) for the first loop. We observe that for the multiplicities considered (x1, x2, x4), more restorations improve FID.

### C.4. Choice of dataset to restore

In all our experiments in the main paper, we perform the restoration of each Dataloop always starting from the same corrupted samples. In this section, we ablate this choice by comparing to restoring from the previous Dataloop's restoration i.e. treating them as corrupted samples at a smaller noise level. Table 9 shows conditional FID of the restored dataset in Loop 2 if we restore from scratch vs continuing from the previous loop (Loop 1), in the case of Blur ($\sigma_B = 0.6$). We

Table 9. Comparison of restoration methods. FID ↓ (lower is better). Loop2 restorations.

| Corruption | | Restore from scratch | Restore from previous loop |
|---|---|---|---|
| Blur | $\sigma_B = 0.6$ | 3.862 | **3.478** |
| | $\sigma_B = 0.8$ | 4.481 | **4.156** |
| JPEG | $q = 25$ | 4.789 | **4.318** |
| | $q = 18$ | 5.260 | **4.678** |

Table 10. Effect of number of posterior samples. Cifar Blur 0.6

| Posterior samples | FID |
|---|---|
| x1 | 4.85 |
| x2 | 4.70 |
| x4 | **4.52** |

observe that restoring from the previous loop's dataset, to the effect of "trusting" the previous loop's model, actually leads to better conditional FID than restoring from scratch. This shows that errors can accumulate even as models get better from one loop to another.

### C.5. Origin of improvement in terms of diffusion times

We also analyze where, in terms of noise times, the improved performance of the Ambient Loops model is coming from compared to the baseline Ambient Omni model. Figure 6 shows average EDM loss curves across different noise times averaged over the entire clean cifar-10 dataset, providing a window of analysis into the per-noise performance of the models. We observe that, for all four corruptions, the Loop1 models are better *for all noise times* than the Omni models. This is initially surprising as Ambient Loops can only facilitate the learning of information present in the dataset (the low-frequencies of the corrupted data), but can do nothing to recover information lost to the corruption (high-frequencies of the corrupted data). The conundrum is explained by the theoretical results: the posterior estimated samples are closer distributionally to the clean samples than the initial blurry samples, and so it is possible to extract more information from them at all noise levels. Indeed, the conditional FID results empirically support this assumption, as seen in 1: the corrupted datasets have conditional FIDs in the 10 to 60 range depending on the severity of the corruption, but the Loop 0 restored datasets all have FIDs below 5.

## D. ImageNet restoration using a powerful model

In this section, we show we can also use the state-of-the-art image-generation model Flux to "restore" ImageNet, i.e., to improve the quality of ImageNet images without compromising the diversity of the dataset. Specifically, we take the low-quality samples definition from Ambient Omni (Daras et al., 2025c), those with CLIP-IQA quality below the 80th percentile, and restore it by first adding noise with $t = 0.3$ ($x_t := (1 - t) \cdot x + t \cdot z$, with $z \sim \mathcal{N}(0, 1)$) and then performing posterior sampling using Flux. As we can see visually in Figure 7, only the quality of the images improved, without altering what is actually depicted in the image. To actually quantify the improvement of quality on ImageNet, we use the CLIP-IQA metric. The results for the original ImageNet and the improved ImageNet are presented in Tables 12 (average CLIP-IQA quality) and 12 (winrate). These metrics, paired with our visuals, show that the quality of the dataset improved.

Table 11. Statistical Summary

| Dataset | Mean ± Std | Min | Max |
|---|---|---|---|
| Original ImageNet | 0.7251 ± 0.1572 | 0.0184 | 0.9959 |
| **Flux-Restored** | **0.7469 ± 0.1501** | **0.0265** | **0.9971** |

Table 12. Winrate Comparison

| Dataset winner | % occurrence |
|---|---|
| Original ImageNet wins | 37.82% |
| **Flux-Restored wins** | **62.17%** |

## E. Experimental Details

### E.1. Training hyperparameters

For all our controlled experiments in Section 5, we use the EDM (Karras et al., 2022) codebase. We provide the full hyperparameters for our models in Table 13.

For our text-to-image experiments, we use the MicroDiffusion (Sehwag et al., 2025) codebase. We start with the checkpoint from the Ambient Diffusion Omni work, available in the following URL: https://huggingface.co/giannisdaras/ambient-o. This model is a Diffusion Transformer (Peebles & Xie, 2023) utilizing Mixture-of-Experts (Shazeer et al., 2017) (MoE) feedforward layers with a total parameter count of $\approx 1.16B$ parameters. We use this checkpoint to restore the DiffDB

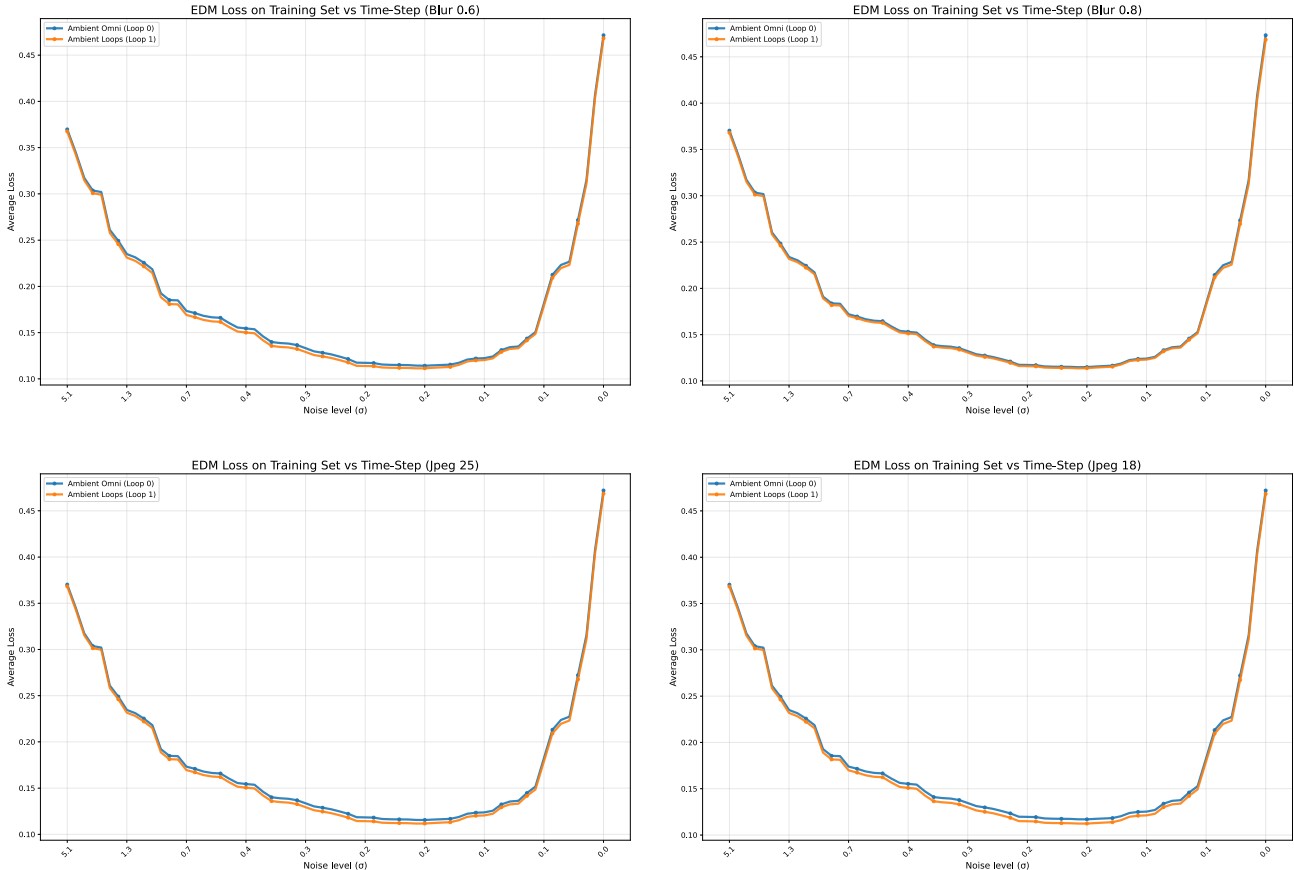

*Figure 6.* EDM loss vs noise level for Ambient Diffusion Omni (Daras et al., 2025c) vs Ambient Loops (Loop 1) models across four corruptions on Cifar-10. Loops models have increased denoising performance *across all* noise levels for all four corruptions.

| Hyperparameter | Value |
|---|---|
| Architecture Type | Diffusion U-net (Song et al., 2020) |
| # model params | 55M |
| Batch size | 512 |
| Maximum training duration (kimg) | 200,000 |
| EMA half-life (kimg) | 500 |
| EMA ramp-up ratio | 0.05 |
| Learning-rate ramp-up (kimg) | 10,000 |

*Table 13.* Training hyperparameters for our controlled experiments on CIFAR-10.

dataset, after we map it to a noise level $\sigma_{\text{DiffusionDB}} = 2.0$ (see Section 5.1). We finetune the Ambient Omni model on the resulting dataset for $\approx$ 10M images (5000 optimization steps with a batch size of 2048). We provide the full hyperparameters in Table 14.

For our protein experiments (see Section 5.3), we use the scaled up Genie-2 architecture that the authors of Ambient Proteins (Daras et al., 2025b) proposed. The hyperparameters used to train this model (and our subsequent model) are available in Table 15. We start with the pre-trained model from (Daras et al., 2025b), that is available in the following URL: https://huggingface.co/jozhang97/ambient-short. This model was trained on a subset of the AFDB (Varadi et al., 2022) dataset, that is available in the following URL: https://huggingface.co/datasets/jozhang97/afdb-tm40. The AFDB dataset contains AlphaFold predictions for protein foldings that vary on accuracy. The Ambient Proteins (Daras et al., 2025b) authors used AlphaFold's self-reported pLDDT score, to map protein structures to a noise level at which the data can be approximately trusted. In particular, the authors trained their model with 1000 discrete noise levels (cosine diffusion

| Hyperparameter | Value |
|---|---|
| **Model Architecture** | |
| VAE | SDXL Base 1.0 |
| Text encoder | DFN5B-CLIP ViT-H/14 |
| Backbone | MicroDiT_XL_2 |
| Latent resolution | 64 |
| Image resolution | 512 |
| Input channels | 4 |
| Precision | bfloat16 |
| Positional interpolation scale | 2.0 |
| $p_{\text{mean}}$ | 0 |
| $p_{\text{std}}$ | 0.6 |
| **Optimization** | |
| Train batch size | 2048 |
| Caption drop probability | 0.1 |
| Optimizer | AdamW |
| Learning rate | $8 \times 10^{-5}$ |
| Weight decay | 0.1 |
| $\epsilon$ | $1 \times 10^{-8}$ |
| Betas | (0.9, 0.999) |
| Gradient clipping (norm) | 0.5 |
| Low-precision LayerNorm | amp_bf16 |
| **Exponential Moving Average (EMA)** | |
| Smoothing | 0.9975 |
| EMA start | 2048 kimg |
| Half-life | None |
| **Training Schedule** | |
| Scheduler | Constant LR |
| Max duration | 10,240 kimg |

*Table 14.* Training hyperparameters for our MicroDiffusion experiment.

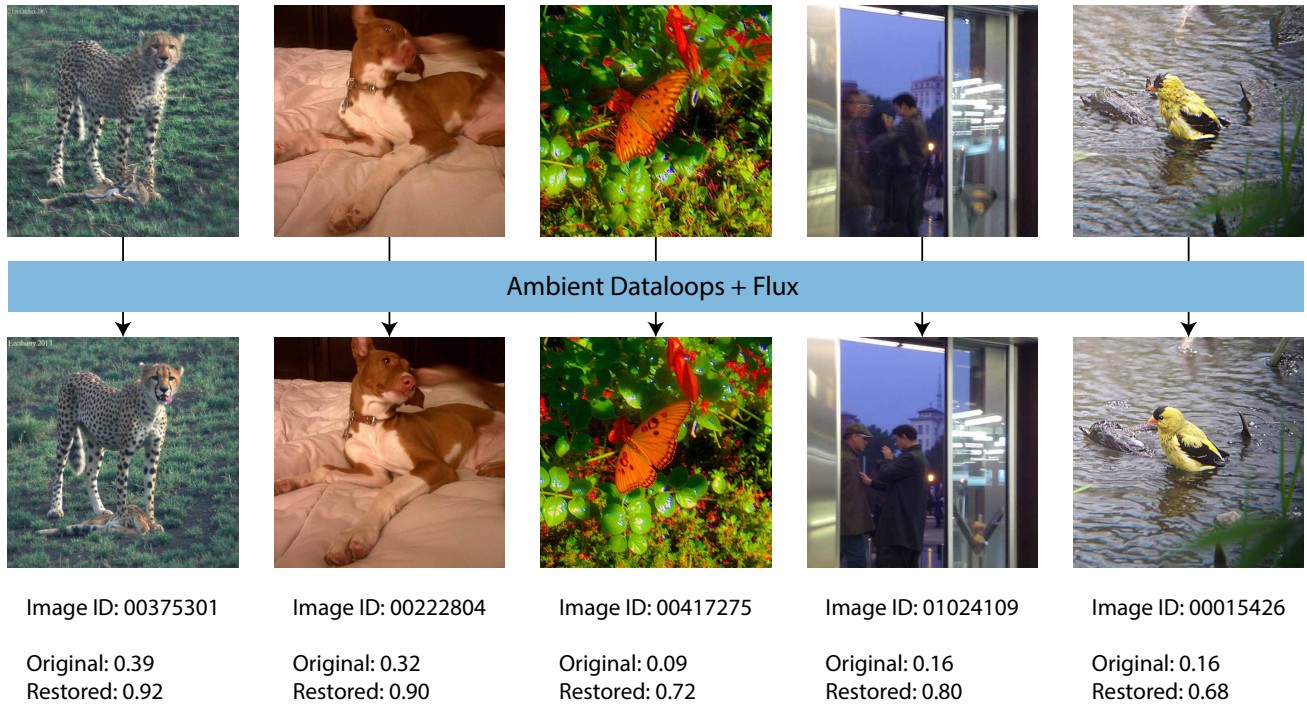

*Figure 7.* Examples of ImageNet images well-restored by Flux

schedule) and the following mapping:

$$\begin{cases} [1, 1000], \ \text{pLDDT} \geq 90 \\ [600, 1000], \ 90 > \text{pLDDT} \geq 80 \\ [900, 1000], \ 80 > \text{pLDDT} \geq 70. \end{cases} \tag{20}$$

This mapping should be interpreted as follows: proteins that have pLDDT$\geq 90$ can be used for all diffusion times. Proteins with pLDDT in $[80, 90)$ can be used only for times $[600, 1000]$. Finally, proteins with pLDDT in $[70, 80)$ can be used only for times $[900, 1000]$.

We use the model from (Daras et al., 2025b) to denoise their training set. This results in a new dataset of protein structures that is used for our subsequent training run. Before we proceed with the training phase, we need to quantify how much the noise of the original dataset was reduced. To do so, we experiment with constant increments in pLDDT, and we find that an assumed $+3$ increment in the pLDDT of each denoised protein yields the optimal results. Hence, if a protein before the denoising had a pLDDT $x$, its new pLDDT after the denoising is assumed to be $x + 3$, and we use the same mapping function (Eq. 20) to convert it to a noise level. With this new dataset and with the training hyperparams detailed in Table 15, we train our model.

### E.2. Training Computational Requirements

We trained all of our models for the image domain on a cluster of $8 \times$H200 GPUs. Each CIFAR training (Section 5 takes a maximum of 1 day on this compute (200K training kimg), but usually less since models trained on corrupted data converge faster. Our MicroDiffusion (Sehwag et al., 2025) runs (Section 5.1) take only $\approx 7$ hours on this compute budget, since we start finetuning from the model of Daras et al. (2025c).

Our protein experiments (Section 5.3) require more compute and we run them on $48 \times$H200 GPUs. The training takes roughly 12 hours on this compute. This is similar to the compute time needed at the Ambient Proteins (Daras et al., 2025b) paper.

*Table 15.* Training hyperparameters for our proteins experiment.

| Hyperparameter | Value |
|---|---|
| ***Diffusion*** | |
| Number of timesteps | 1,000 |
| Noise schedule | Cosine |
| ***Model Architecture*** | |
| Single feature dimension | 384 |
| Pair feature dimension | 128 |
| Pair transform layers | 8 |
| Triangle dropout | 0.25 |
| Structure layers | 8 |
| ***Training*** | |
| Optimizer | AdamW |
| Number of training proteins | 196k |
| Number epochs | 200 |
| Warmup iterations | 1,000 |
| Total batch size | 384 |
| Learning rate | $1.0 \times 10^{-4}$ |
| Weight decay | 0.05 |
| Minimum protein length | 20 |
| Maximum protein length | 256 |
| Minimum mean pLDDT | 70 |

### E.3. Restoration Computational Requirements

To run each loop of our algorithm, we need to perform a restoration (i.e. denoising) of the existing dataset. For our CIFAR-10 experiments, the denoising takes $\leq 10$ minutes on $8\times$ H200 GPUs. This number refers to a full denoising and it is usually faster since we only need to do a partial denoising each time. Restoration time becomes significant for our MicroDiffusion experiments since the dataset is much larger. In particular, restoring the whole Diffusion DB requires 4 days on $48\times$H200 GPUs. In practice, we only run it for 1 day on the same hardware since the finetuning stage only runs for $1/4$th of an epoch and hence we only need to restore $1/4$th of DiffusionDB. The restoration cost is also significant for our protein results. Restoration takes roughly 28 hours on $48\times$H200 GPUs (compared to the 12 hours needed for training). The reason for this overhead is that denoising with posterior sampling requires multiple steps for a single datapoint. This overhead can be reduced by using one step or few-step variants of diffusion models, but we leave this direction for future work.

### E.4. Evaluation pipeline

In this subsection we provide some details regarding our evaluation pipeline.

**Controlled Experiments.** All our controlled experiments are performed on CIFAR-10. CIFAR-10 consists of $50,000$ images at $32 \times 32$ resolution. For all our experiments, we keep $10\%$ of the training set untouched (i.e. 5000 clean images) and we synthetically corrupt the rest using the different corruption models studied in the paper.

When we compute FID, we always compute it *with respect to the full uncorrupted CIFAR-10*. This allows to truly access whether the model learned the clean distribution and not the distribution after the corruption. To compute FID, we follow standard practice and we generate $50,000$ images with the trained model. To estimate the statistical significance of the results, we compute FID 3 times starting with a different generation seed each time. The standard deviations that are reported in Table 1 capture the variability across different FID computations. To find the best obtained FID, we evaluate different checkpoints throughout the training and we report results for the checkpoint that obtained the best mean FID across the three different seeds.

Regarding sampling hyperparameters, we adopt the sampling scheme of the EDM (Karras et al., 2022) paper and we keep

all the sampling parameters fixed across different experiments. In particular, we use a sampling budget of 35 Network Function Evaluations (NFEs) and we follow a deterministic sampler using the second order Heun's discretization method. It is possible that additional benefits could be obtained by tuning separately the sampling hyperparameters for each run.

**Text-to-image experiments.** For our text-to-image results, we report zero-shot FID on the COCO2014 validation split, following the evaluation protocol of Sehwag et al. (2025). In particular, we generate 30K thousand images using random prompts from the COCO 2014 validation set and we measure the distribution similarity between real and synthetic samples using FID. During our sampling, we use classifier-freee guidance (Ho & Salimans, 2022), i.e., our denoising prediction is formed using a linear combination of the unconditional and conditional prediction of the model, as follows:

$$\hat{x}_0 = (1 + w)h_\theta(x_t, t, c) - wh_\theta(x_t, t, \emptyset), \tag{21}$$

where $w$ controls the guidance strength. We fix $w = 1.5$ for our COCO evaluations. Our sampling budget is 30 NFEs and, similar to our controlled experiments on CIFAR, we use a second-order deterministic sampler based on Heun's method.

**Statistical Significance of reported FIDs.** To ensure that our FID improvements in the text-to-image experiments are statistically significant, we compute FID with three different sets of seeds and we report the mean and the standard deviation in Table 16.

| Model | FID Mean | FID Std |
|---|---|---|
| Ambient Omni | 10.79 | 0.04 |
| Ambient Loops | 10.11 | 0.03 |

*Table 16.* Mean FID scores and standard deviations for our text-to-image experiments computed over three different sets of seeds.

Beyond the variations due to the sampling seeds, there is also variation across training runs. To understand the magnitude of that variation, we re-trained a text-to-image model using a different initial set of random weights and we obtained an FID of 10.14 instead of the 10.11 Table 16 reports. Since this result is within the standard deviation of the sampling seed variation, we do not perform any more training results.

# F. Noisy dataset formation

Throughout the paper, we assumed access to some dataset $\mathcal{D}_0 = \{(x_{t_i}, t_i)\}_{i=1}^N$ of datapoints $x_{t_i}$ corrupted with additive gaussian noise of variance $\sigma^2(t_i)$. However, in our experiments we deal with points that have been corrupted in various forms beyond additive gaussian noise. In this section of the appendix, we clarify how we transform a dataset of arbitrarily corrupted points into a dataset of points corrupted with additive gaussian noise. The methodology described here is from the paper Ambient Diffusion Omni (Daras et al., 2025c). We summarize the main points of the framework here to ensure our paper self-contained. The actual setting studied is the following: there is a dataset $D_h = \{x_{0i}\}_{i=1}^{n_1}$ of high-quality points $x_{0i}$ sampled according to some target distribution $p_0$. There is also a dataset $D_l = \{y_{0i}\}_{i=1}^{n_2}$ of potentially low-quality or out-of-distribution points sampled from some other distribution $q_0$. For notation purposes, we use the r.v. $X_t$ to denote a point from $p_t$ and $Y_t$ to denote a point from $q_t$. Ambient Omni uses $D_h$ as a reference set in order to annotate the points in $D_l$ i.e. to assign a diffusion time $t_i$ to each point $y_{0i}$ from $D_l$. In particular, the dataset $D_0$ that is assumed in our setting is formed by concatenating the dataset $D_h$ with a noisy version of $D_l$ denoted by $D_l' = \{(y_{0i} + \sigma(t_i)Z, t_i)\}_{i=1}^{n_2}$ where $Z \sim \mathcal{N}(0, I_d)$. The rest of the discussion focuses on how these annotation noise levels $t_i$ are decided. The authors of (Daras et al., 2025c) propose two ways perform the annotation:

- Fixed sigma: With this method, all the points in $D_l$ are assigned to the same noise level $\sigma(t_n)$. This noise level is selected such that the TV distance $\mathrm{TV}(p_{t_n}, q_{t_n}) < \epsilon$ for some threshold $\epsilon > 0$. In practice, the value $\sigma(t_n)$ is tuned as a hyper-parameter to achieve optimal validation performance. This value can also be estimated usign a classifier, as explained below.

- Time-dependent classifier: In this method, each point in $D_l$ is assigned its own noise level $\sigma(t_i)$. To do so, we first train a time-dependent classifier (Dhariwal & Nichol, 2021) $c_\theta^\star(W_t, t)$ that tries to estimate wether a point $W_t$ came

from the distribution $p_t$ (label 1) or $q_t$ (label 0), where the r.v.

$$W_t = \begin{cases} X_t & \text{with probability } 0.5 \\ Y_t & \text{with probability } 0.5 \end{cases}$$

. The classifier is trained with cross-entropy loss trying to predict 1 for points coming from $p_t$ and 0 for points coming from $q_t$. Simply put, this classifier is trained to distinguish between points from the high-quality and low-quality distributions under noise. Once this classifier is trained, we can use it to annotate points in $D_l$. In particular, we can assign to each point the minimal diffusion time $t_i$ for which the classifier becomes approximately confused. Formally, for a point $y_{0i}$ in $D_l$ we assign the time $t_i = \inf_t \mathbf{E}_{Z \sim N(0, I_d)}[c_\theta^\star(y_{0i} + \sigma(t)Z, t)] >= \frac{1}{2} - \epsilon$. This classifier can also be used to assign a noise level for the entire dataset (see the above fixed sigma section) by finding the minimal noise level $t_n$ for which the validation accuracy of the classifier is $\leq \frac{1}{2} + \epsilon$.

To avoid the complexities of training classifiers, we use the fixed sigma method for all experiments. However, all of our findings naturally extend to the case of per-data point noise levels.

Now that the set-up has been explained, we need to clarify where do these sets $D_h$ and $D_l$ come from in our experiments, and how the fixed sigma value is decided. For our controlled experiments on Cifar-10, $D_h$ is simply a random 10% of the dataset, and $D_l$ is the remaining 90% but artificially corrupted with either blurring or JPEG compression. The fixed sigma value is obtained by doing hyper-parameter search; the small size of cifar permits such extensive optimization. For our text-to-image experiments, $D_h$ contains all the images from the contextual captions, segment anything, and journeyDB datasets, while $D_l$ contains the images form the diffusionDB dataset. The fixed sigma value is set to 2.0; this value is taken directly from the experiments of Ambient Omni (Daras et al., 2025c). For our proteins experiments, we can use AlphaFold to assess the quality of each protein in our training set. AlphaFold reports a self-confidence score, pLDDT, that we use as a proxy of quality. $D_h$ contains all the proteins with pLDDT $> 90$; these proteins are considered our high-quality data. Proteins with PLDDT $< 90$ are grouped into three low-quality sets; the noise level for each one of them is assigned using the fixed sigma method and hyper-parameter tuning. Please refer to Section G for details about this annotation scheme.

# G. Proteins Appendix

## G.1. Metrics

Backbone-only generative protein models are principally evaluated with two metrics: designability and diversity.

Designability measures the quality of generated proteins. 100 backbones each of lengths 50, 100, 150, 200, and 250 are generated by the model. To assess whether these backbones could actually be made by some sequence, ProteinMPNN (Dauparas et al., 2022) is used to generate eight candidate amino acid sequences per backbone. These are then folded back into structures using ESMFold (Lin et al., 2023), and if any of these is sufficiently close to the original backbone (RMSD $< 2$ Å), the backbone is considered designable. Designability is then defined as the percentage of generated backbones which are designable.

Diversity measures whether a set of generated structures is highly redundant or if it contains a wide array of meaningfully different proteins. To evaluate diversity, Foldseek is used to cluster the set of designable backbones with a TM-score threshold of 0.5. Diversity is defined as:

$$\text{Diversity} = \frac{\text{Number of Clusters}}{\text{Number of Designable Proteins}}$$

In practice, designability and diversity are at odds. Maximizing diversity typically requires generating less likely structures, some of which will not be designable.

*Table 17.* **Designability and diversity for protein structure generation.**

| Model | Designability (%↑) | Diversity (↑) |
|---|---|---|
| *Ambient Proteins* (L0, $\gamma = 0.35$) | **99.2** | 0.615 |
| **Ambient Loops** (L1, $\gamma = 0.35$) | 99.0 | **0.703** |
| Proteina (FS $\gamma = 0.35$) | 98.2 | 0.49 |
| Genie2 | 95.2 | 0.59 |
| FoldFlow (base) | 96.6 | 0.20 |
| FoldFlow (stoc.) | 97.0 | 0.25 |
| FoldFlow (OT) | 97.2 | 0.37 |
| FrameFlow | 88.6 | 0.53 |
| RFDiffusion | 94.4 | 0.46 |
| Proteus | 94.2 | 0.22 |

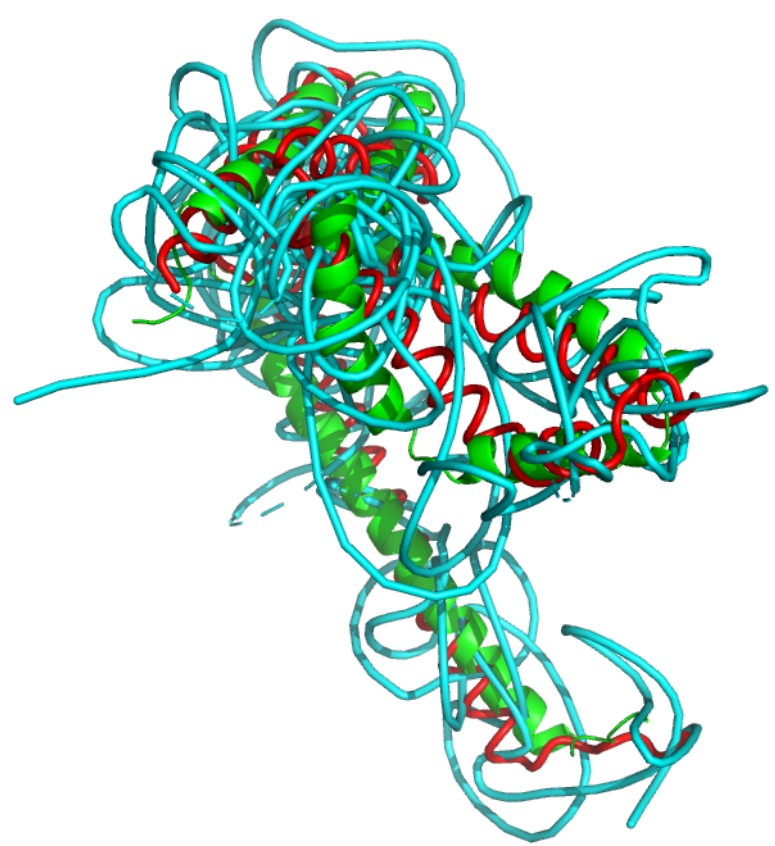

*Figure 8.* Example denoising (red) of a noisy version (cyan) of the green protein.

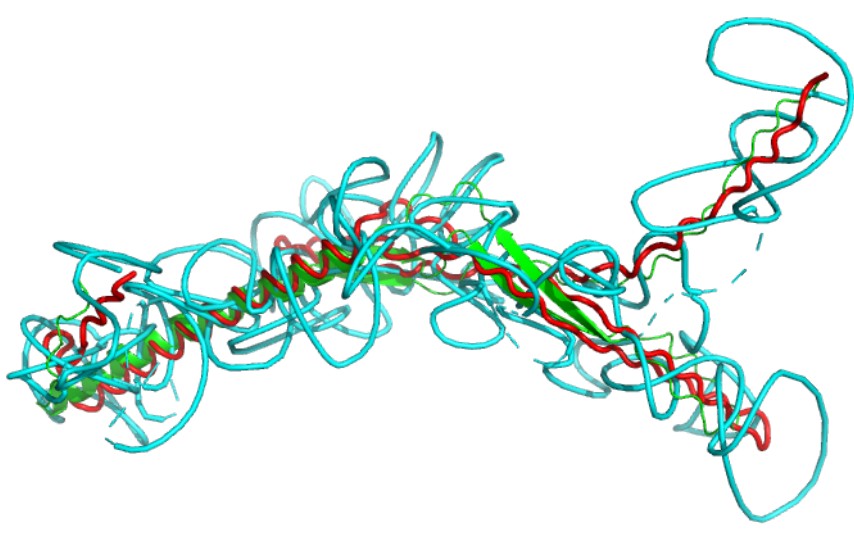

*Figure 9.* Example denoising (red) of a noisy version (cyan) of the green protein.

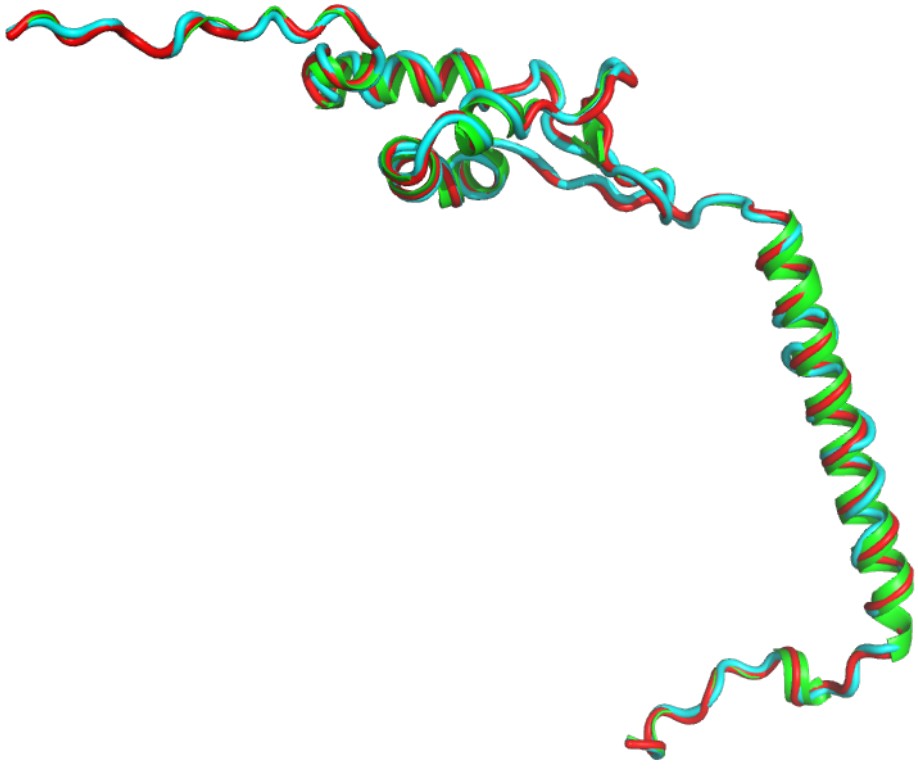

*Figure 10.* Example denoising (red) of a noisy version (cyan) of the green protein.

## G.2. Additional Results

# H. Synthetic 2D experiments

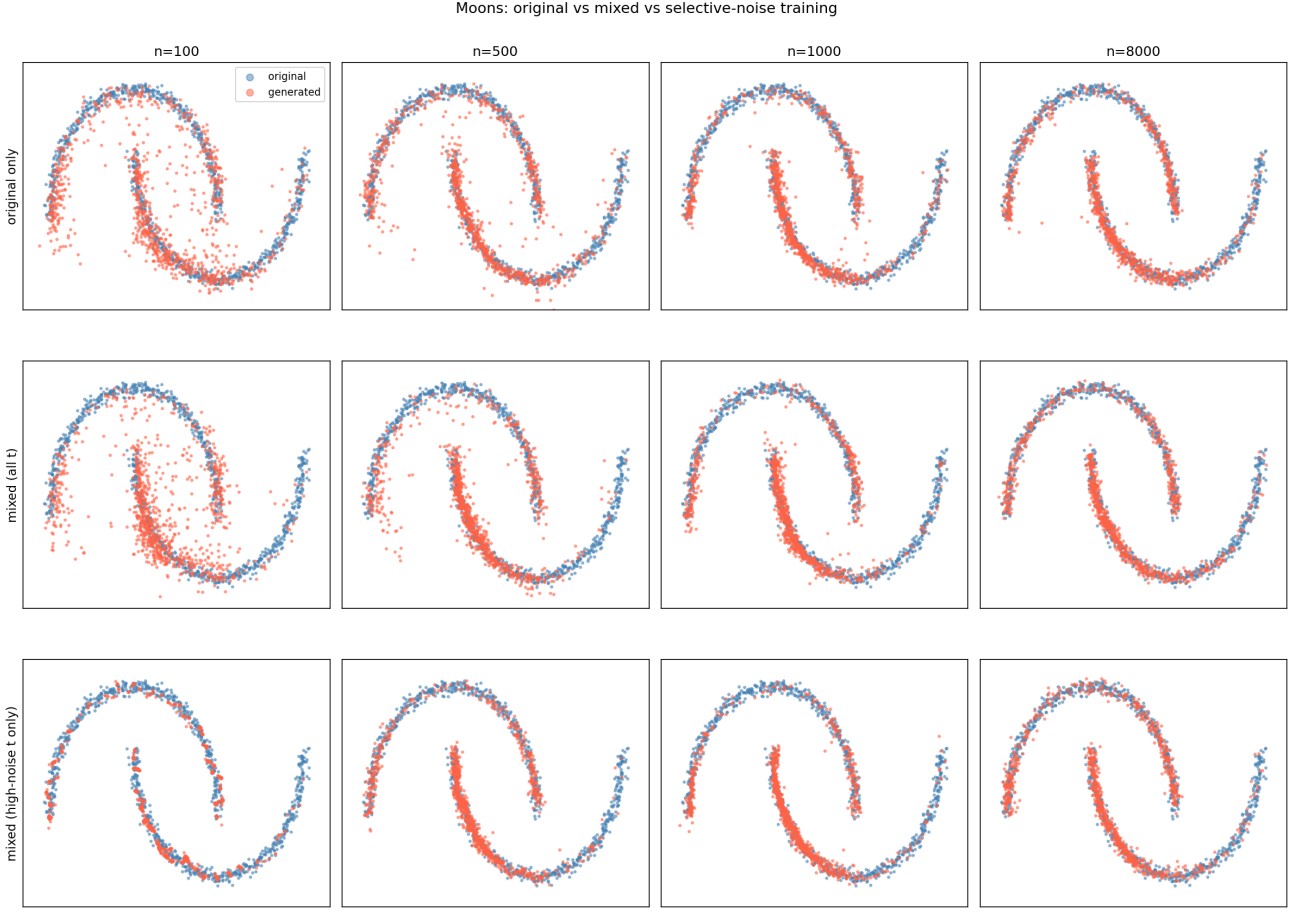

*Figure 11.* Ambient Dataloops on Synthetic 2D data

Figure 11 shows the results of applying our method on synthetic 2D data. In the first row, we depict (with red) the estimation you get to the ground truth distribution (blue) when you use different number of training points $n \in \{100, 500, 1000, 8000\}$.

As shown, the initial estimation is rough for a low number of training points. We use those poorly trained models to generate $8000 - n$ synthetic datapoints and we train new models on the combination of the original $n$ real points and $8000 - n$ synthetic points. The results are shown in Row 2. As shown, the estimation is still pretty bad because the synthetic data is not perfect. On Row 3, we perform the Loops idea. In particular, we train on both the real and the synthetic points, but the synthetic ones are further corrupted by additive Gaussian noise and used only for high-diffusion times. This leads to a remarkable improvement in estimation on this 2-D dataset, providing compelling evidence and intuition for our approach.

