# OpenReview forum: "Ambient Dataloops: Generative Models for Dataset Refinement"
_ICML.cc/2026/Conference — ICML 2026 regular_

### Official Review · Reviewer_5YWw · 2026-03-09

**Soundness:** 3
**Presentation:** 4
**Significance:** 3
**Originality:** 2
**Overall Recommendation:** 5
**Confidence:** 4

**Summary:**

In this paper, the authors propose Ambient Dataloops, which extends the prior method Ambient Diffusion into an iterative training loop that progressively refines corrupted datasets using the gradually improved model. Specifically, the model is trained on a mixture of high-quality and low-quality data. To better utilize the low-quality data, the authors first train a model on this combined dataset using Ambient Diffusion, and then use the trained model to refine the low-quality data to a higher quality level. Technically, they make the diffusion model corruption-aware, enabling it to mitigate the negative effects introduced by low-quality data. Experimentally, their method shows better performance than vanilla Ambient Diffusion on unconditional and text-to-image generation tasks, as well as on a protein design task. Overall, the paper provides insight into how low-quality data can be progressively improved through interaction with the model, forming a data–model co-evolution process.

**Compliance With Llm Reviewing Policy:**

Affirmed.

**Final Justification:**

The additional experiments and justifications strengthen both the method and the conclusions. Overall, I believe the authors have addressed my concerns well, and I would like to increase my score to 5.

**Key Questions For Authors:**

* What is the performance (FID) if the model has access to the full original CIFAR training data? How large is the gap between this upper bound and the performance of your method?

**Limitations:**

The method assumes that we know which part the high-quality data is and which part the low-quality data is. However, in practice, we may not know the exact partition. How can your method be applied in that situation?

**Strengths And Weaknesses:**

Strengths:
* The method achieves good performance on a combined dataset containing both high-quality and low-quality data, which reflects a realistic setup in real-world applications. The strong performance across different tasks and datasets validates the robustness and effectiveness of the proposed method.
* I like the clear structure of the paper, as well as the good writing. It makes the ideas and motivations easy for readers to understand.
* The experiments are comprehensive, covering multiple tasks and including ablation studies that analyze the effects of hyperparameters such as the number of loops.

Weaknesses:
* The theoretical part is somewhat weak. A major part of the theoretical analysis relies on the assumption that the true score function is learned in the first round. Under this assumption, the conclusion that diffusion can push the distribution q_t toward p_t appears rather straightforward, since it essentially assumes a well-approximated score function. Although the authors provide a brief analysis of the learning error, they do not present a thorough analysis of how this error propagates through the iterative loop. In practice, even a small error could be amplified after several loops. The performance degradation observed after many loops in Figure 3 may also be related to this issue.
* I think an experiment on a synthetic dataset would further demonstrate how the low-quality data shifts from q_t to the high-quality distribution p_t. [1] provides several toy 2-D distributions, such as moons and circles. Adding visualizations showing how the corrupted datasets are gradually refined on these 2-D distributions would be an interesting and illustrative example. Compared to Figure 1 in your paper, synthetic 2-D datasets have a known ground-truth distribution, which provides a more straightforward way to demonstrate the gradual refinement process.
[1]: https://github.com/tanelp/tiny-diffusion
* Model collapse is a closely related problem that the authors also discuss in the paper. I recommend adding the following recent literature:
[2]: Hugo Cui, Cengiz Pehlevan, and Yue M. Lu. “A precise asymptotic analysis of learning diffusion models: theory and insights”. arxiv preprint arxiv:2501.03937 (2025).
[3]: Lianghe Shi, Meng Wu, Huijie Zhang, Zekai Zhang, Molei Tao, Qing Qu. "A Closer Look at Model Collapse: From a Generalization-to-Memorization Perspective". NeurIPS, 2025.
[4]: Shi Fu, Yingjie Wang, Yuzhu Chen, Xinmei Tian, and Dacheng Tao. “A Theoretical Perspective: How to Prevent Model Collapse in Self-consuming Training Loops”. ICLR, 2025.

---

> ### Author Rebuttal · Authors · 2026-03-31
>
> We thank the Reviewer for their detailed and thoughtful Review. In what follows, we address the Reviewer's comments.
>
> **(1) Theoretical assumption on having access to the perfect score**
>
> We completely agree with the Reviewer that the theoretical results are weakened by the reliance on the perfect score function for the reconstruction step. As it turns out, we can completely get rid of this reliance by repeating the exact proof methodology of Lemma 4.2, but by also accounting for an error in the score being used for the reconstruction.
>
> This error leads to an extra additive term in the KL distance between the improved $q_t$ distribution and the desired $p_t$ one.
> In particular, before we had that:
>
> $D_{KL}(f(q_t), p_t) \leq (1-C) D_{KL}(q_t, p_t)$
> where $f$ is the operator that contracts $q_t$ towards $p_t$ using the perfect score.
>
> Using the same proof technique, but by accounting for $f$ using a score with expected squared error upper-bounded by $\epsilon$, we get:
>
> $D_{KL}(f(q_t), p_t) \leq (1-C_1) D_{KL}(q_t, p_t) + C_2 \epsilon$.
>
> The full formal result is stated below, and its proof is the same as before:
>
> https://imgur.com/a/4fsWsI1
>
>
> This means that if by doing Omni on Loop 0 we can get a score that has some error $\epsilon$, we can still use this score to contract the distance of the $q_t$ samples, but up to a certain saturation point; the distance cannot be made smaller than $C_2\epsilon$.
>
> This is intuitively correct, and it perfectly aligns with the Reviewer's comment that *"The performance degradation observed after many loops in Figure 3 may also be related to this issue."*
>
> We will include the updated Lemma, proof (that stays pretty much the same other than carrying this error term in the calculations) and the discussion in the Camera Ready version of our work. We believe that this addition significantly strengthens our theoretical argument and provides justification for the success of our experimental validation across all these diverse settings and the saturation of it after a certain limit of iterations.
>
>
> **(2)** Synthetic experiments on 2-D data.
>
> **We followed the Reviewer's recommendation, and we got a super interesting result on 2-D data that we plan to include in our Figure 1**!
>
> The result is shown below:
>
> https://imgur.com/a/wFnHc6x
>
> In the first row, we depict (with red) the estimation you get to the ground truth distribution (blue) when you use different number of training points $n \in \{100, 500, 1000, 8000\}$.
>
> As shown, the initial estimation is rough for a low number of training points. We use those poorly trained models to generate $8000 - n$ synthetic datapoints and we train new models on the combination of the original $n$ real points and $8000 - n$ synthetic points. The results are shown in Row 2. As shown, the estimation is still pretty bad because the synthetic data is not perfect. On Row 3, we perform the Loops idea. In particular, we train on both the real and the synthetic points, but the synthetic ones are further corrupted by additive Gaussian noise and used only for high-diffusion times. This leads to a remarkable improvement in estimation on this 2-D dataset, providing compelling evidence and intuition for our approach.
>
> We deeply thank the Reviewer for suggesting this experiment and for helping us improve our work.
>
> **(3) Missing citations**
>
> We commit to adding citations to all the papers the Reviewer proposed and to discuss them extensively in our Camera Ready version.
>
> **(4) Ground truth CIFAR-10 number**
>
> The unconditional FID on the full uncorrupted CIFAR-10 using the EDM codebase (which we rely on) is $1.98$. There is still a large gap compared to the results we get in Table 1 for Blur/JPEG corruptions (around FID 5-6), but our method still achieves a 17% improvement in FID in some cases compared to Ambient Diffusion Omni (which was the previous state-of-the-art for this problem).
>
> **(5) Determining the good and the bad set in practice**
>
> The Reviewer makes a fair point that the starting point of our method is a good set and a bad set. Doing this splitting might not be trivial in practice. This issue is discussed in the Ambient Diffusion Omni paper, which we build upon.  To summarize, there are two important points: (A) we only need to be sure about points that belong to the "good" set as this defined the target distribution and (B) the "good" set doesn't have to be large, in fact it can be an order of magnitude smaller than the "bad" set without a significant impact on the performance.
>
> That separation of the "best-quality" samples can be done by using proxy reward models (e.g. Aesthetic Score, Pickscore). These models have many imperfections, but for the outliers (best images) they are typically pretty reliable.
>
> In the absence of such proxies, one can ask human annotators or perform small-scale training experiments on different data subsets as done in the Data Mixing literature to find the best samples (e.g. see https://arxiv.org/abs/2403.16952).

---

> > ### Author Rebuttal · Reviewer_5YWw · 2026-04-04
> >
> > Thanks for the response. The additional experiments and justifications strengthen both the method and the conclusions. Overall, I believe the authors have addressed my concerns well, and I would like to increase my score to 5.

---

### Official Review · Reviewer_YMtt · 2026-03-11

**Soundness:** 4
**Presentation:** 3
**Significance:** 4
**Originality:** 3
**Overall Recommendation:** 5
**Confidence:** 4

**Summary:**

This paper proposes Ambient Dataloops, a training framework for diffusion models on datasets that contain low-quality samples. Ambient Dataloops introduces a co-evolution process between the dataset and the model, where the dataset is refined through generation and the model is subsequently trained on the generated data. Previous approaches typically address low-quality data by filtering or removing such samples from the dataset. However, this strategy may reduce the diversity of the dataset. Instead, the proposed method aims to improve the dataset itself using a generative model rather than removing data. Furthermore, training generative models on generated data may lead to error amplification, known as the self-consuming loop problem. To mitigate this issue, the proposed method treats generated samples as noisy observations during training.

**Compliance With Llm Reviewing Policy:**

Affirmed.

**Final Justification:**

After carefully reviewing the paper, the authors' rebuttal, and the other reviewers' comments, I maintain my positive assessment and recommend acceptance for this paper.

I find the contributions of this paper to be highly significant and original. When training diffusion models on datasets with low-quality samples, the conventional strategy of filtering out data often compromises dataset diversity. This paper introduces a refreshing methodology that restores and improves the dataset itself, preserving diversity. Furthermore, the proposed approach elegantly addresses the critical issue of bias amplification (the "self-consuming loop") when using generated data. By treating generated samples as noisy observations, the authors establish a safe and effective co-evolution process between the dataset and the model. The soundness of this work is also a major strength, supported by solid theoretical analysis and comprehensive empirical results on practical domains like text-to-image and protein structure generation.

In the rebuttal, the authors thoughtfully addressed my primary empirical concern: the lack of a baseline experiment where images generated by Ambient Omni are used directly for training. I had pointed out that while this naive approach would likely trigger a self-consuming loop and degrade accuracy, confirming this behavior experimentally was essential to fully guarantee the paper's core contribution. The authors provided these exact baseline results, which clearly demonstrated the anticipated degradation and strongly highlighted the necessity and efficacy of their proposed approach. This addition completely resolved my concerns and solidified the paper's claims.

Overall, the rebuttal reinforced my prior assessment. Given the strong theoretical foundation, innovative approach to dataset restoration, and the strengthened empirical validation provided during the rebuttal, I believe this paper presents a highly valuable contribution to the generative modeling community at ICML.

**Key Questions For Authors:**

Q1. Could the authors provide experimental results where images generated by Ambient Omni are directly used for training? Based on the self-consuming loop phenomenon, such a setting may lead to degradation in generation performance. However, verifying this baseline in the current experimental setting would be important for establishing the contribution of the proposed method.

**Limitations:**

yes

**Strengths And Weaknesses:**

Strengths
- Dataset restoration for real-world datasets: In training diffusion models on datasets containing low-quality samples, a common approach is to filter out low-quality data. However, this strategy may reduce dataset diversity. This paper instead proposes an approach that improves the dataset itself rather than removing data, introducing a new methodology for training on low-quality datasets.
- Avoiding the self-consuming loop: Prior work has pointed out that training generative models using generated data can lead to bias amplification, known as the self-consuming loop problem. The proposed method mitigates this issue by treating generated samples as noisy observations. This design enables generated data to be used safely for training generative models and allows a co-evolution process between the dataset and the model.
- Theoretical analysis: The paper provides a theoretical analysis showing that the co-evolution process between the dataset and the model can improve model training.
- Experiments considering practical applications: In addition to experiments under synthetic noise conditions, the authors conduct experiments on text-to-image generation and protein structure generation to evaluate the method in more practical settings.

Weaknesses
- Repeated text in the manuscript: The same sentence appears twice consecutively in Lines 74–80.
- Parameter tuning in Dataset Restoration: As mentioned in the paper, parameter tuning in the Dataset Restoration process is complex. This could also become a limitation when considering practical applications.
- Limited novelty compared to Ambient-Omni: Part of the contribution, namely the ability to train with noisy data, relies on the Ambient-Omni framework, which makes the novelty relative to Ambient-Omni somewhat limited.

---

> ### Author Rebuttal · Authors · 2026-03-31
>
> We appreciate the Reviewer’s time and thoughtful review.
>
> **(1) Typos**
>
> We thank the Reviewer for spotting the duplicated text in our submission; we will fix it in the Camera Ready version, and we apologize for the typo.
>
> **(2) Novelty**
>
> Regarding the novelty comment of the Reviewer, we underline that the novelty in this work lies in the idea of dataset-model co-evolution (recursive application of Ambient Omni, where the noise level is reduced gradually by using the model itself to denoise the dataset) and the theoretical analysis for the proposed framework. That said, we acknowledge that our contributions rely on the foundation of the Ambient Omni work.
>
> **(3) Additional Experiments**
>
> The Reviewer asks for “experimental results where images generated by Ambient Omni are directly used for training”.
>
> This experiment is already present in Appendix Table 4, page 17, last row (trust rate infinity).
> As shown, the optimal unconditional FID that can be obtained by this method is 4.99 (at Loop 2), and doing further loops leads to collapse, e.g., the FID at Loop 4 is 8.93. By using our method that treats the generated data as noisy, we can obtain an unconditional FID of 4.56 in that setting.
>
> We agree with the Reviewer that this is an important baseline, and we will highlight it more prominently in the Camera Ready version of our work.
>
> **(4) Parameter tuning**
>
> Based on the Reviewer’s point, we ran additional analysis on Table 4 to provide guidance on how to select restoration parameters: (A) when to stop, and (B) the rate at which you need to denoise the dataset until you reach the stopping point. Regarding (B), the denoising rate, it turns out that the conditional FID of the first Loop is a highly predictive metric for the best noise level schedule for subsequent loops.
>
> | | Uncond FID | Cond FID |
> |---|---|---|
> | Loop 1 → Loop 2 | 0.82 | **0.82** |
> | Loop 2 → Loop 3 | 0.54 | **1.00** |
> | Loop 3 → Loop 4 | 0.36 | **0.96** |
> | Loop 1 → Best Uncond | 0.71 | **0.93** |
>
> The above table reports spearman rank correlations for Unconditional FID of the next loop with the Unconditional and Conditional FIDs of the previous loop. We observe that the conditional FID of loop $i$ is a remarkably strong predictor of unconditional FID at loop $i+1$ (ρ = 0.82–1.0), much stronger than unconditional FID predicting itself at the next loop (ρ drops from 0.82 to 0.36). Most importantly, conditional FID at loop 1 also strongly predicts the best unconditional FID achieved across all loops (ρ = 0.93), showing its usefulness for determining which trust rate will ultimately perform best.
>
> The best Loop1 conditional FID is achieved by ρ=2^3 with 3.98, and has best unconditional FID of 4.65 over all the loops. The absolute best unconditional FID is achieved by ρ=2^2 at 4.56, which is less than a 0.1 difference.
>
> In the absence of a reference set to compute conditional FID with, reward models such as Aesthetic Score (Schuhmann et al. https://github.com/LAION-AI/aesthetic-predictor) for image quality and Vendi Score for diversity (Friedman et al. https://arxiv.org/abs/2210.02410) may be used as conditional metrics to determine the best restored dataset without held-out data.
>
> This reduces the hyperparameter search of the denoising rate to a sweep only for the first loop. That can be done efficiently via binary search (as the rate->FID function is convex).
>
> | Trust rate ρ (noise level) | Loop 0 Uncond. FID | Loop 0 Cond. FID | Loop 1 Uncond. FID | Loop 1 Cond. FID | Loop 2 Uncond. FID | Loop 2 Cond. FID | Loop 3 Uncond. FID | Loop 3 Cond. FID | Loop 4 Uncond. FID | Loop 4 Cond. FID |
> | :--- | :--- | :--- | :--- | :--- | :--- | :--- | :--- | :--- | :--- | :--- |
> | 2^0.25 | 5.67 | 4.46 | 5.44 | 4.46 | 5.14 | 4.14 | 5.34 | 4.16 | 5.49 | 4.18 |
> | 2^0.5 | 5.67 | 4.46 | 5.21 | 4.26 | 5.3 | 4.1 | 5.26 | 4 | 5.1 | 4 |
> | 2^1 | 5.67 | 4.46 | 5.17 | 4.21 | 5.31 | 4.08 | 5.22 | 3.91 | 4.96 | 3.61 |
> | 2^2 | 5.67 | 4.46 | 5.14 | **4.03** (top-2 loop1 cond.) | 4.77 | 3.53 | **4.56** (top-1 uncond.) | 3.76 | 5.08 | 5.09 |
> | 2^3 | 5.67 | 4.46 | 4.88 | **3.98** (top-1 loop1 cond.) | 4.74 | 3.72 | **4.65** (top-2 uncond.) | 4.25 | 5.58 | 5.68 |
> | 2^4 | 5.67 | 4.46 | 4.9 | 4.04 | 4.76 | 3.98 | 4.66 | 5.19 | 6.13 | 8.02 |
> | Infinity | 5.67 | 4.46 | 5.03 | 4.14 | 4.99 | 4.86 | 5.86 | 7.92 | 8.93 | 12.69 |
>
> Regarding A) when to stop, given the above discussion on B) finding the rate, we have reduced it to stopping when conditional metrics become worse compared to the previous loop. This again can be done through use of reward models such as Aesthetic Score (Schuhmann et al. https://github.com/LAION-AI/aesthetic-predictor) or ImageReward (Xu et al. https://arxiv.org/abs/2304.05977) which don't require held-out data.

---

> > ### Author Rebuttal · Reviewer_YMtt · 2026-04-03
> >
> > Thank you for your efforts during the rebuttal phase. I keep my positive recommendation.

---

### Official Review · Reviewer_VVzY · 2026-03-12

**Soundness:** 4
**Presentation:** 4
**Significance:** 3
**Originality:** 3
**Overall Recommendation:** 4
**Confidence:** 3

**Summary:**

In this paper, the authors propose a method for training diffusion models using a mixture of clean and low-quality data. The key idea is the recursive application of ambient diffusion, allowing the dataset and model to co-evolve.

**Compliance With Llm Reviewing Policy:**

Affirmed.

**Key Questions For Authors:**

See weakness.

**Limitations:**

yes

**Strengths And Weaknesses:**

**Strengths**

* The motivation is sound. In many cases, we have access to a large amount of low-quality data and only a small amount of high-quality data.
* The idea is conceptually simple—just recursively applying ambient diffusion. The implementation should also be fairly straightforward.

**Weaknesses**

1. In Table 4, it seems that regardless of the chosen noise level, the FID increases at loop 4. This suggests that there are cases in which the method helps and cases in which it does not. Do we have a theoretical or empirical way to predict whether it will lead to improvement?

2. Related to point 1, the two hyperparameters—the number of loops and the noise level for each loop—seem difficult to tune. In practice, high-quality held-out data may not be available (this is essentially what the paper is trying to solve). How can these parameters be tuned without access to an oracle metric?

---

> ### Author Rebuttal · Authors · 2026-03-31
>
> We thank the Reviewer for their time and thoughtful review. In what follows, we address the Reviewer’s questions regarding the selection of the hyperparameters of our method.
>
> The Reviewer mentions: “In Table 4, it seems that regardless of the chosen noise level, the FID increases at loop 4”. We want to clarify that this is not true since there is a noise level (x-axis point) for which Loop 4 gives the optimal result in terms of FID in Figure 3.
>
> That said, as the Reviewer correctly points out, the performance of our method depends on two hyperparameters: (A) when to stop, (B) the rate at which you need to denoise the dataset until you reach the stopping point. Miscalibrating either of these two parameters will lead to suboptimal performance.
>
> Following the Reviewer’s recommendation, we ran additional analysis on Table 4 to provide guidance on how to select these hyperparameters. Regarding (B), the denoising rate, it turns out that the conditional FID of the first Loop is a highly predictive metric for the best noise level schedule for subsequent loops.
>
> | | Uncond FID | Cond FID |
> |---|---|---|
> | Loop 1 → Loop 2 | 0.82 | **0.82** |
> | Loop 2 → Loop 3 | 0.54 | **1.00** |
> | Loop 3 → Loop 4 | 0.36 | **0.96** |
> | Loop 1 → Best Uncond | 0.71 | **0.93** |
>
> The above table reports spearman rank correlations for Unconditional FID of the next loop with the Unconditional and Conditional FIDs of the previous loop. We observe that the conditional FID of loop $i$ is a remarkably strong predictor of unconditional FID at loop $i+1$ (ρ = 0.82–1.0), much stronger than unconditional FID predicting itself at the next loop (ρ drops from 0.82 to 0.36). Most importantly, conditional FID at loop 1 also strongly predicts the best unconditional FID achieved across all loops (ρ = 0.93), showing its usefulness for determining which trust rate will ultimately perform best.
>
> The best Loop1 conditional FID is achieved by ρ=2^3 with 3.98, and has best unconditional FID of 4.65 over all the loops. The absolute best unconditional FID is achieved by ρ=2^2 at 4.56, which is less than a 0.1 difference.
>
> In the absence of a reference set to compute conditional FID with, reward models such as Aesthetic Score (Schuhmann et al. https://github.com/LAION-AI/aesthetic-predictor) for image quality and Vendi Score for diversity (Friedman et al. https://arxiv.org/abs/2210.02410) may be used as conditional metrics to determine the best restored dataset without held-out data. The problem of validating ML models without held-out sets is present in many subject areas and other works, particularly with regard to maximising human preferences (Jiang et al. https://arxiv.org/abs/2406.11191, Xu et al. https://arxiv.org/abs/2304.05977), and is not unique to us.
>
> This reduces the hyperparameter search of the denoising rate to a sweep only for the first loop. That can be done efficiently via binary search (as the rate->FID function is convex).
>
> | Trust rate ρ (noise level) | Loop 0 Uncond. FID | Loop 0 Cond. FID | Loop 1 Uncond. FID | Loop 1 Cond. FID | Loop 2 Uncond. FID | Loop 2 Cond. FID | Loop 3 Uncond. FID | Loop 3 Cond. FID | Loop 4 Uncond. FID | Loop 4 Cond. FID |
> | :--- | :--- | :--- | :--- | :--- | :--- | :--- | :--- | :--- | :--- | :--- |
> | 2^0.25 | 5.67 | 4.46 | 5.44 | 4.46 | 5.14 | 4.14 | 5.34 | 4.16 | 5.49 | 4.18 |
> | 2^0.5 | 5.67 | 4.46 | 5.21 | 4.26 | 5.3 | 4.1 | 5.26 | 4 | 5.1 | 4 |
> | 2^1 | 5.67 | 4.46 | 5.17 | 4.21 | 5.31 | 4.08 | 5.22 | 3.91 | 4.96 | 3.61 |
> | 2^2 | 5.67 | 4.46 | 5.14 | **4.03** (top-2 loop1 cond.) | 4.77 | 3.53 | **4.56** (top-1 uncond.) | 3.76 | 5.08 | 5.09 |
> | 2^3 | 5.67 | 4.46 | 4.88 | **3.98** (top-1 loop1 cond.) | 4.74 | 3.72 | **4.65** (top-2 uncond.) | 4.25 | 5.58 | 5.68 |
> | 2^4 | 5.67 | 4.46 | 4.9 | 4.04 | 4.76 | 3.98 | 4.66 | 5.19 | 6.13 | 8.02 |
> | Infinity | 5.67 | 4.46 | 5.03 | 4.14 | 4.99 | 4.86 | 5.86 | 7.92 | 8.93 | 12.69 |
>
> Regarding A) when to stop, given the above discussion on B) finding the rate, we have reduced it to stopping when conditional metrics become worse compared to the previous loop. This again can be done through use of reward models such as Aesthetic Score (Schuhmann et al. https://github.com/LAION-AI/aesthetic-predictor) or ImageReward (Xu et al. https://arxiv.org/abs/2304.05977) which don't require held-out data.

---

> > ### Author Rebuttal · Reviewer_VVzY · 2026-04-04
> >
> > Using surrogate quality metric sounds like a reasonable patch. I still recommend acceptance.

---

> > > ### Author Response · Authors · 2026-04-06
> > >
> > > We thank the reviewer for their insightful comments and help at refining our discussion regarding parameter tuning in practice. Since they believe their concerns have been fully resolved and that surrogate quality metrics are a reasonable patch, we would appreciate it if the Reviewer would consider increasing their score to signal their strong support in our work. If there is anything further we can do to improve our work we remain committed to doing so.

---

### Official Review · Reviewer_Chij · 2026-03-13

**Soundness:** 2
**Presentation:** 2
**Significance:** 3
**Originality:** 1
**Overall Recommendation:** 4
**Confidence:** 4

**Summary:**

The authors propose an "dataloop" method to address the issue of synthetic data autophagy (where models trained on synthetic data progressively diverge) by restricting synthetic data training to high-noise regimes. The paper includes theoretical analysis to support the approach and evaluates the method across image and protein modalities.

**Compliance With Llm Reviewing Policy:**

Affirmed.

**Final Justification:**

The authors did a good job to fix certain weaknesses. I still not on par with some aspects of the papers, like the low geneval scores. Especially, improving some rewards like image rewards usuallly correlate with better genevals scores.
I'm not as strongly against accepting this paper, but if the paper is accepted, i would like the authors to take the time to adress some of this weaknesses

**Key Questions For Authors:**

1. **Empirical Degradation:** How do you reconcile the claim that this method "solves" synthetic data autophagy with the data in Figure 3, which clearly shows the model entering the MADness regime (degrading FID) by Generation 4?
2. **Missing Baselines (SIMS):** How does your high-noise training approach compare empirically to guidance-based methods like SIMS, which directly push the generation away from the synthetic manifold?
3. **ImageNet Evaluation:** Can you provide results on ImageNet 256 to prove the method generalizes beyond the easily overfitted CIFAR dataset?

**Limitations:**

The paper fails to adequately discuss the clear empirical limitations of the method, most notably its inability to prevent generational degradation (MADness) past Generation 3. Furthermore, the reliance on toy datasets like CIFAR and the omission of state-of-the-art data-efficient T2I baselines severely restrict the claims that can be made regarding the method's real-world scalability and utility.

**Strengths And Weaknesses:**

**Strengths:**
* **Crucial Problem Space:** Addressing the synthetic data autophagy issue is critical for the future scalability of generative models.
* **Simplicity:** The proposed method is straightforward and computationally simple to implement.
* **Theoretical Grounding:** The inclusion of theoretical analysis provides a helpful framework for understanding the high-noise regime approach.
* **Modality Diversity:** Evaluating the method on protein generation in addition to standard image generation is a strong and highly appreciated addition.

**Weaknesses:**
* **Empirical Failure:** Based on the reported results, the approach does not actually prevent autophagy. The model enters the "MADness" (Model Autophagy Disorder) regime after just four generations. According to Figure 3, the FID of Generation 4 is worse than both Generation 2 and Generation 3, directly contradicting the claim that the method "solves" the issue.
* **Negligible T2I Improvements:** The gains in Text-to-Image (T2I) generation are virtually non-existent. The GenEval score remains stagnant at 47, and the 0.5 improvement in FID is well within the standard error margins for the metric.
* **Inadequate Evaluation Datasets:** The reliance on CIFAR for empirical validation is unconvincing, as most modern diffusion networks simply overfit on CIFAR. Evaluations on ImageNet 256 are required to properly demonstrate the method's efficacy in pixel space.
* **Missing Foundational Citations:** The literature review is severely lacking. The authors discuss the "MADness" phenomenon without citing the foundational paper that coined it. Furthermore, they miss recent, highly relevant works that tackle this exact problem using guidance, such as SIMS. Missing citations include:
    * Alemohammad, S., et al. (2024). **Self-Consuming Generative Models Go MAD.** *ICLR 2024.*
    * Alemohammad, S., et al. (2025). **Self-Improving Diffusion Models with Synthetic Data (SIMS).** *ICLR 2025.*
* **Missing Baselines:** The experimental setup ignores critical baselines:
    * **Synthetic Data Baselines:** There is no comparison against SIMS, which proposes a robust solution by guiding away from synthetic data.
    * **T2I Baselines:** The paper fails to compare against similarly sized networks  including Stable Diffusion 1.5, Stable Diffusion 2.1 or highly data-efficient models like PixArt-$\alpha$, and recent works like:
        * Dufour, N., et al. (2024). **Don't Drop Your Samples! Coherence-Aware Training Benefits Conditional Diffusion.** *CVPR 2024.*
        * Degeorge, L., et al. (2025). **How far can we go with ImageNet for Text-to-Image generation?**
* **Poor Absolute Performance:** The reported GenEval score of 47 is exceptionally low. For context, "How far can we go with ImageNet" achieves a GenEval score of 60 using a much smaller 350M parameter network trained on only 1.2M images.
* **Suboptimal Experimental Design:** Rather than demonstrating marginal improvements on weak setups, a much more compelling experiment would be to start from a highly efficient, minimal-data baseline (e.g., the ImageNet-only setup from Degeorge et al.) and progressively inject synthetic data to measure quality improvements.

---

> ### Author Rebuttal · Authors · 2026-03-31
>
> **(1)** **The reviewer wrongly claims that we missed foundational citations**. The papers mentioned **are all cited** in our submission. See Lines 563-568.
>
> **(2) On preventing MADness**
>
> We start by making the trivial statement that MADness can be entirely prevented by our method if one does not attempt to denoise the dataset beyond a certain limit. No change in the dataset means identical training and hence no MADness. Of course, it is not always possible to know (A) when to stop, (B) at what rate you need to denoise the dataset until you reach the stopping point. Miscalibrating either of these two parameters [(A), (B)] will lead to madness.
>
> Figure 3 in our paper studies this. The fundamental denoising limit for this dataset is $\approx$ 0.03 (this is parameter (A)), and one good way to reach this parameter is by following the green line schedule (this is parameter (B)). Additionally, we point out that Loop 4 is **not** always worse than other loops: there is an x point in our Figure 3 for which Loop 4 is strictly better than all other loops.
>
> That said, we will clarify that miscalibration of parameters (A), (B) can lead to madness within our framework.
> We further performed an analysis of how these parameters can be chosen without sweeping (see response to Reviewer VVzY for details).
>
> We also want to point out that for our method, performance always improves before it becomes worse. Recursive self-training used to be considered harmful. In our work, we show that if done right, it can be helpful.
>
> **(3) Statistical significance of our T2I results**
>
> The Reviewer claims that the *“0.5 improvement in FID is well within the standard error margins for the metric”*. We report the FID mean and standard deviation for our method and the baseline on three different sets of seeds for the COCO dataset:
>
> |Model|FID Mean|FID Std|
> |:---|:---|:---|
> |Ambien Omni|10.7893|0.0397|
> |Ambient Loops|10.1083|0.0321|
>
> The difference **is statistically significant**.
>
> **(4) Poor performance on GenEval**
>
> GenEval is a benchmark designed to measure T2I alignment. Our method does not improve alignment to text, but the quality of the generated images. This is why performance on GenEval is stagnant compared to the non-looped baseline, but improved in terms of FID.
>
> The Reviewer is further concerned that our baseline (MicroDiffusion) is not relevant because it has low absolute performance on GenEval compared to other baselines, such as the “How far can we go with ImageNet for T2I generation?” baseline.
>
> We point out that even though the model the Reviewer suggested has better T2I alignment on the specific prompts of GenEval, it appears to be a weaker model (than both MicroDiffusion and Loops) in other benchmarks.
>
> |Model|PickScore|Aes.Score|HPSv2|ImageReward|
> |:---|:---|:---|:---|:---|
> |Loops Micro.|**21.22**|5.11|**0.2520**|**0.4225**|
> |ImgNet base|20.94|5.46|0.24|0.2|
> |ImgNet+L-POP|21.04|**5.67**|0.25|0.24|
>
> Visual inspection confirms the superior quality of our model.
> Further, the ImageNet model is incapable of generating fantastical prompts and scenes not found within ImageNet.
>
> **Prompts:**
> (i) A tennis field,
> (ii) An astronaut riding a unicorn,
> (iii) A ferocious fire-breathing dragon
>
> **Images:**
> https://imgur.com/a/WRpWyL2
>
>
> **(5) Comparison with SIMS**
>
> We initially skipped this comparison because the setting is different.
> The starting point of SIMS is an uncorrupted dataset while our method starts with a corrupted one. This is why SIMS guides away from the generated data and towards the dataset while we try to transcend the input dataset.
>
> That said, following the Reviewer’s suggestion, we perform the following experiment:
>
> (a) We use a MicroDiffusion model trained on all the datasets, including the lower-quality synthetic DiffDB dataset.
> (b) We use a MicroDiffusion model trained without the low-quality synthetic DiffDB dataset.
>
> We use SIMS to guide towards (b) and away from (a). The results in terms of quality and COCO FID are shown below:
>
> |Omega|COCO FID|CLIP-IQA|
> |:---|:---|:---|
> |0.0|13.03|0.4171|
> |0.05|13.05|0.4204|
> |0.1|13.04|0.4210|
> |0.25|13.08|0.4207|
> |0.5|13.23|0.4213|
>
> SIMS increases quality but decreases diversity, leading to worse FID. Our method improves both.
>
> **(6) Additional experiments on ImageNet**
>
> The Reviewer asks for additional experiments on ImageNet. We start with the model that the Reviewer suggested, and we used it to improve the bottom 20% of ImageNet in terms of quality.
> Below, we report evaluations that confirm that the model improved it:
>
> | Metric | Original | Restored |
> | :--- | :--- | :--- |
> | Aesthetic | 4.03±.34 | **4.41**±.48 |
> | HPSv2 | 0.20±.03 | **0.23**±.03 |
> | ImgReward | -0.23±.88 | **0.24**±.89 |
> | PickScore | 20.39±1.25 | **20.70**±1.21 |
>
> Finetuning the baseline on this restored data improves 3 of 4 metrics:
>
> |Model|PickScore|Aes.Score|HPSv2|ImageReward|
> |:---|:---|:---|:---|:---|
> |Base|20.07|**4.85**| 0.22 |-0.37|
> |Loops|**20.13**|4.81|**0.24**|**-0.19**|

---

> > ### Author Rebuttal · Reviewer_Chij · 2026-04-04
> >
> > 1)   First, I sincerely apologize to the authors for the oversight regarding the references. I acknowledge that the two references I previously mentioned are indeed included in the paper.
> >
> >  2)   Regarding the prevention of MAD, the core issue is maintaining stability during training on synthetic data. Implementing a stopping criterion does not inherently demonstrate that MAD has been prevented. Furthermore, several prior works, including the authors baseline MicroDiffusion, have already established that incorporating synthetic data can improve training across specific metrics.
> >
> > 3)    Please clarify if the reported standard deviations are computed using inference redundancy. Since the majority of uncertainty stems from the training process, a more rigorous approach would require training multiple networks to properly capture this variance.
> >
> > 4)    The authors appear to dismiss GenEval, which is a central metric for evaluating Text-to-Image improvements, and the proposed method does not demonstrate gains on it. Showing progress on aesthetic rewards through iterative training on synthetic data loops is unconvincing. Aesthetic metrics can remain artificially high even when the distribution completely collapses, a phenomenon that naturally occurs with MAD. While I agree that ImageNet T2I is not an ideal baseline when evaluated strictly on aesthetics, this is largely due to the nature of the ImageNet data itself. Demonstrating improved aesthetic metrics on the ImageNet T2I dataset using the proposed method would be significantly more compelling, as it would prove the method can induce performance gains not inherently present in the initial data.
> >
> > 5)    I appreciate the inclusion of these new results, which provide an important point of comparison. It would also be valuable to see an evaluation of the authors' method combined with SIMS.
> >
> >  6)   Could the authors confirm if these results are based on ImageNet class-conditional generation? Furthermore, why is FID not reported? Aesthetic metrics are not standard practice for evaluating ImageNet T2I models.
> >
> > An additional is that the authors seem to insists that they care more about aesthetics than other metrics. If that's the case, the authors should then compare to alignment methods that also base their studies on this metrics. Furthermore, one could argue that GRPO based alignment methods are in a way already doing dataloops, since they train on their own synthetic data.

---

> > > ### Author Response · Authors · 2026-04-06
> > >
> > > We agree with the reviewer that other works have explored the promise of synthetic data. However, there is a difference in the setting: in MicroDiffusion, the synthetic data is generated by a stronger teacher, while in our paper it is refined by our model itself. Moreover, the framework we have introduced can be used to improve *any kind* of data, including synthetic data, as well as highly compressed, blurry, and other low-quality images.
> > >
> > > Next, it is not standard in Diffusion Model literature to report training variances because they are very expensive runs, so the original rebuttal result of 10.11 $\pm$ 0.03 was obtained using inference redundancy. However, to satisfy the reviewer’s request we did one more training run of our Ambient Loops model with a different seed and obtained very similar results: FID=10.14, compared to our original inference redundancy result of FID=10.11
> > >
> > > We also do not disregard GenEval, we just clarify that GenEval measures text-to-image alignment, and our method does not target text-to-image alignment. Our method improves the standard FID metric and other metrics such as aesthetic reward quality. The model does not collapse because FID becomes lower. Further, the aesthetic metrics we used to evaluate ImageNet models are the exact same as the paper the reviewer suggests as a critical baseline (Table 7 / Section 4, page 8 ), and our method performs better. Lastly, of these aesthetic metrics PickScore, HPSv2, and ImageReward are all prompt conditional. Hence, if the distribution had indeed collapsed, these metrics would also be severely penalised and we do not observe this.
> > >
> > > To demonstrate gains not present in the initial data, we report aesthetic metrics on a random prompt subset of the 80% of ImageNet T2I data whose corresponding images we did not improve using our method. As seen below, the Loops model improves on 3/4 aesthetic metrics:
> > >
> > > | Model | PickScore | Aes.Score | HPSv2 | ImageReward |
> > > | :--- | :--- | :--- | :--- | :--- |
> > > | Base | 20.85 | **5.16** | 0.25 | 0.50 |
> > > | Loops | **20.90** | 5.10 | **0.27** | **0.59** |
> > >
> > > All the ImageNet results in the rebuttal are obtained with text-conditional generation, using exactly the same generation pipeline and prompts as the original “How far can we go with ImageNet for Text-to-Image generation?” the reviewer suggested for comparison.
> > >
> > > We did not initially report FID metrics because they measure the distance of a model’s generations to a specific reference dataset (the one used for FID calculation). Since our method changes the training dataset by improving it, this makes FID on the original training dataset an inadequate metric. An amazing high-quality diverse model would score poorly on FID when evaluated against a low-quality dataset. To satisfy the reviewer’s request, we perform two additional tests with FID. One on the 80% of ImageNet T2I prompts whose images we did not improve with our method, and another on COCO as the “How far can we go with ImageNet for Text-to-Image generation?” paper also does. As seen below, our Loops model beats the baseline in both cases. Moreover, the COCO FID score of ImageNet models are significantly worse (>20) than that of MicroDiffusion models.
> > >
> > > | Model | 80% ImageNet train FID | COCO FID |
> > > | :--- | :--- | :--- |
> > > | Base | 7.99 | 20.46 |
> > >  | Loops | **7.41** | **19.39** |
> > >
> > > Finally, we don’t compare with RL algorithms because these can be run on top of our method and are orthogonal contributions. We have a method to improve pre-training and finetuning of diffusion models, and RL is something you can run on top if you have a reward model you trust. Moreover, optimizing directly for a reward often leads to “reward hacking” because rewards (especially ML ones) are imperfect, but using it as an evaluation (and not as optimization target) is a reasonable practice that the community uses. The same principle holds for FID. You can optimize for it and get bad outputs that give low FID. The community doesn’t do this but still uses it as an evaluation method.
> > >
> > > The Reviewer currently holds a rating of 2 (clear reject) despite the numerous experiments we did during the rebuttal that addressed some of the reviewer’s concerns and in contrast to the strong positive assessment of all the other reviewers. In this latest reply, we did our best to provide evidence that addresses any remaining concerns and ultimately improve our paper. We thank the Reviewer for their time and effort and we would appreciate it if they would consider reevaluating their assessment in the light of this evidence. We commit to including these experiments and discussions in the camera ready version of our work.

---

### Decision · Program_Chairs · 2026-04-30

**Decision:**

Accept (regular)

**Comment:**

The paper proposes Ambient Dataloops, an iterative framework for training diffusion models on mixed real and synthetic data training by progressively restoring low-quality samples while avoiding self-consuming model collapse through noise-aware training. Reviewers viewed the problem as important and timely, and highlighted the method’s simplicity, practicality, theoretical support, and various applications including image, text-to-image, and protein generations.

The main concerns were about the limits of the empirical claims, sensitivity to hyperparameter tuning, the strength of the theory under imperfect score estimation, and positioning relative to prior work. The rebuttal strengthened the paper by clarifying related work, adding analyses on hyperparameter selection and error propagation, and providing additional empirical evidence, including stronger baselines and ImageNet-style evaluations. Several reviewers stated that their concerns were fully resolved after the rebuttal.

Overall, I recommend acceptance. The paper makes a useful contribution on training generative models from imperfect data while preserving diversity, and its remaining limitations do not outweigh its practical and conceptual value.